# Estimation of stationary and non-stationary moving average processes in the correlation domain

**Martin Dodek** *, Eva Miklovičová

Institute of Robotics and Cybernetics, Faculty of Electrical Engineering and Information Technology Slovak University of Technology in Bratislava, Bratislava, Slovakia

* martin.dodek@stuba.sk

**Data Availability Statement:** All relevant data are within the paper and its Supporting information files.

**Funding:** This work was supported by the call for PhD students and young researchers of Slovak

## Abstract

This paper introduces a novel approach for the offline estimation of stationary moving average processes, further extending it to efficient online estimation of non-stationary processes. The novelty lies in a unique technique to solve the autocorrelation function matching problem leveraging that the autocorrelation function of a colored noise is equal to the autocorrelation function of the coefficients of the moving average process. This enables the derivation of a system of nonlinear equations to be solved for estimating the model parameters. Unlike conventional methods, this approach uses the Newton-Raphson and Levenberg–Marquardt algorithms to efficiently find the solution. A key finding is the demonstration of multiple symmetrical solutions and the provision of necessary conditions for solution feasibility. In the non-stationary case, the estimation complexity is notably reduced, resulting in a triangular system of linear equations solvable by backward substitution. For online parameter estimation of non-stationary processes, a new recursive formula is introduced to update the sample autocorrelation function, integrating exponential forgetting of older samples to enable parameter adaptation. Numerical experiments confirm the method's effectiveness and evaluate its performance compared to existing techniques.

## Introduction

Estimation of the parameters of moving average processes, which means determining a moving average model that best fits the colored noise sequence, obtained from an experiment is one of the fundamental problems in statistics and signal processing. The methods developed for this theoretical problem can potentially be applied also in the system and control theory, signal processing, economics and financial modelling and forecasting [1–5]. Despite seeming relatively straightforward, this problem is more complicated than it might appear. The main reason is that the sequence of input noise signal is always unknown in practical scenarios. Therefore, the standard prediction-error approaches based on minimizing the least squares criterion cannot be directly applied in the case of moving average stochastic processes because the input white noise is unknown [6]. Instead, special modifications such as the pseudolinear

University of Technology in Bratislava to start a research career (Grant 23-03-01-B Impulsive Control of Biosystems). Call for PhD students and young researchers was supported by call 09I03-03-V05, financed by RRP SR. The funders had no role in study design, data collection and analysis, decision to publish, or preparation of the manuscript.

**Competing interests:** The authors have declared that no competing interests exist.

regression approach or the iterative extended least squares method have to be used. However, these have their specific limitations and drawbacks.

Concerning the review of other approaches to estimate the moving average processes, the following relevant methods can be found in the literature.

The classical prediction error method to estimate the moving average processes is the two-step Durbin's method [7, 8]. The first step consists of fitting an autoregressive model to the measured output noise sequence via the ordinary least squares method. In the second step, the estimated autoregressive model is used to determine the parameters of the moving average process according to the Yule-Walker equations. A different variant of the Durbin's method uses the least squares estimator also for the second step of the identification. In that case, the estimate of the input white noise sequence is determined by filtering the output noise sequence by the inverse of the estimated autoregressive model. The second step considers this estimated input noise to form the standard linear regression problem for the moving average model and to estimate its parameters directly by the least squares method. However, this method is known to provide biased estimates under some circumstances while not being designed for online estimation in the case of non-stationary processes.

Another popular strategy is the pseudolinear regression [6], which considers the fact that the input noise is unmeasurable, so it is replaced by its estimate based on the model prediction error. However, thus estimated input noise is dependent on the estimated parameters themselves, which ultimately causes adverse nonlinear effects, hence the name pseudolinear regression. The pseudolinear regression problem can be solved by the extended least squares iterative algorithm [9] or it can also be solved online by its recursive version called Recursive Pseudo Linear Regression (abbr. RPLR) [6, 10, 11]. However, a practical limitation of the RPLR is that it assumes a monic polynomial of the model, i.e the first coefficient is always one, what makes impossible to estimate general models that involve variable noise levels.

From the class of correlation-based methods, in [12–14], the problem of moving average process estimation from the sample autocovariance was formulated as a semidefinite program making the minimization of the autocovariance fitting criterion a convex optimization problem. To obtain the estimate of the model coefficients from the solution of the autocovariance fitting problem, a spectral factorization algorithm was used. This approach was reported to have an accuracy comparable to maximum likelihood estimators, and yet it can be obtained as the solution of a convex optimization problem in a polynomial time. However, despite the problem's convexity and improvements in efficiency, the method still requires a significant amount of computational resources as the complexity if more than quadratic. More importantly, this method is designed only for offline estimation of stationary processes without a potential for its extension to non-stationary processes.

Concerning the maximum likelihood methods, which rely on a multidimensional parameter search, it is worth noting that these methods typically necessitate the use of iterative numerical algorithms with relatively high computational complexity (more than quadratic). These algorithms are computationally intensive due to the need to repeatedly evaluate the likelihood function and its derivatives across multiple dimensions. Moreover, maximum likelihood estimation requires making assumptions about the distribution of the output signal, leading to a highly nonlinear optimization problem. These assumptions often involve the probability density function of the underlying data, and any deviation from these assumptions can significantly affect the accuracy of the estimation. There is also a substantial risk of maximum likelihood methods becoming trapped in a local maximum of the likelihood function, particularly when dealing with experimental data. The aforementioned factors significantly hinder the application of maximum likelihood methods for online estimation of non-stationary processes.

It can be concluded that the only information available for identification is the sequence of output colored (correlated) noise and certain assumptions on the statistical properties of the input signal, particularly the whiteness and zero mean.

This paper addresses the problem of moving average processes estimation in the correlation domain rather than in the time domain as in the case of prediction-error methods. One of the motivations for pursuing the approach presented in this paper, compared to other reviewed methods, is its recursive formulation of the parameter estimation problem. This allows for an efficient update of the parameter estimate with each new sample of correlated noise. Additionally, it minimizes the computational complexity of online estimation to the solution of a triangular system of linear equations, eliminates the nonlinear effects typical of pseudolinear regression and enables the estimation of the first model coefficient, which is not possible within the prediction-error methods.

The presented generalization of the identification method to non-stationary processes means that the estimated model will adapt due to the actual parameter changes of the moving average process over time.

The most significant contributions and highlights of this paper can be summarized as follows:

- A new method for offline estimation of stationary moving average processes based on the autocorrelation function matching technique using numerical solution of the corresponding system of nonlinear equations.

- A direct extension of this method to online estimation of non-stationary moving average processes.

- Significantly reduced complexity of the estimation problem in the case of non-stationary processes resulting in a single nonlinear equation and a system of linear equations to be solved each iteration.

- The corresponding linear equation system can be solved effectively by backward substitution due to its upper triangular structure.

- A new recursive formula for efficient updating of the exponentially weighted sample auto-correlation function.

## Materials and methods

### Moving average process

Consider the stationary moving average process defined as [1, 15, 16]

$$\epsilon_{(k)} = g(z)\eta_{(k)} \ , \tag{1}$$

where $\eta \sim \mathcal{N}(0, \sigma_\eta^2)$ is the uncorrelated zero-mean white noise input and $\epsilon$ represents the correlated colored noise output.

The input noise $\eta_{(k)}$ is assumed to be ergodic and to satisfy

$$
\begin{aligned}
E\{\eta_{(k)}\} &= 0 \quad \forall k \ , \\
E\left\{\eta_{(k)}^2\right\} &= \sigma_\eta^2 \quad \forall k \ , \\
E\{\eta_{(k)}\eta_{(k+i)}\} &= 0 \quad \forall k \, , \forall i \neq 0 \ .
\end{aligned}
\tag{2}
$$

Polynomial $g(z)$ in this $n_g$-th order model can be expanded as

$$g(z) = g_0 + g_1 z^{-1} + g_2 z^{-2} + \ldots g_{n_g} z^{-n_g} , \tag{3}$$

where $z^{-n} \eta_{(k)} = \eta_{(k-n)}$ is the backward time-shift operator. Notice the non-unit $g_0$, which scales the instantaneous effect of the input noise as $g_0 \eta_{(k)}$, hence the polynomial in Eq (3) is not monic.

The equivalent difference equation for Eq (1) can be written as

$$\epsilon_{(k)} = \sum_{i=0}^{n_g} g_i \eta_{(k-i)} . \tag{4}$$

The parameter vector $\mathbf{g} \in \mathbb{R}^{n_g+1}$ representing the coefficients $g_i \in \mathbb{R}$ to be estimated gets

$$\mathbf{g} = \begin{bmatrix} g_0 & g_1 & g_2 & \cdots & g_{n_g} \end{bmatrix}^{\mathrm{T}} . \tag{5}$$

In the case of a non-stationary process, the coefficients $g_i$ are time-varying with respect to the sample number $k$ as

$$\epsilon_{(k)} = g(z, k) \eta_{(k)} . \tag{6}$$

The difference equation given by Eq (4) gets

$$\epsilon_{(k)} = \sum_{i=0}^{n_g} g_i(k) \eta_{(k-i)} . \tag{7}$$

In practice, the input noise signal $\eta$ is umeasurable, but it can be estimated based on the inverse filtering of the output $\epsilon$ according to the difference equation given by Eq (4) as

$$\hat{\eta}_{(k)} = \frac{1}{g_0} \left( \epsilon_{(k)} - \sum_{i=1}^{n_g} g_i \hat{\eta}_{(k-i)} \right) . \tag{8}$$

Note, that Eq (8) can be evaluated only considering the estimated model coefficients $g_i$, hence the input noise can never be fully reconstructed.

It is a known fact that estimating moving average processes is more difficult and less straightforward than estimating autoregressive processes [8, 17]. Because the input white noise signal $\eta$ is unmeasurable in practice, the traditional approach based on the least squares minimization of the model single-step-ahead prediction error cannot be directly used to identify the parameters of model given by Eq (1). Therefore, the coefficient vector defined by Eq (5) has to be estimated using only the available finite sequence (time series) of the colored noise signal $\epsilon$ while assuming that the input $\eta$ has the properties of zero-mean white noise as defined in Eq (2). To estimate the coefficient vector of the moving average process model, a novel autocorrelation function matching method will be proposed.

## Autocorrelation function of the colored noise

As the first necessary step, the autocorrelation function of the colored noise $\epsilon$ will be derived. The autocorrelation function $R_{\epsilon\epsilon}(n) : \mathbb{Z} \to \mathbb{R}$ of the stationary colored noise $\epsilon_{(k)}$ is for the lag argument $n \in \mathbb{Z}$ defined by the expectancy [18–20]

$$R_{\epsilon\epsilon}(n) = E\{\epsilon_{(k)} \epsilon_{(k-n)}\} , \tag{9}$$

which holds $\forall k \in \mathbb{N}$.

In the case of a non-stationary process, $R_{\epsilon\epsilon}(n, k)$ will be dependent on the sample number $k$ as

$$R_{\epsilon\epsilon}(n, k) = E\{\epsilon_{(k)}\epsilon_{(k-n)}\} \ . \tag{10}$$

**Stationary process.** Substituting $\epsilon_{(k)}$ and $\epsilon_{(k+n)}$ in the terms of stationary process defined by Eq (4) into Eq (9) yields

$$R_{\epsilon\epsilon}(n) = E\left\{\sum_{i=0}^{n_g} g_i\eta_{(k-i)} \sum_{j=0}^{n_g} g_j\eta_{(k-j-n)}\right\} \ . \tag{11}$$

By expanding the product of summations in Eq (11) we obtain

$$\begin{aligned} R_{\epsilon\epsilon}(n) \ &= E\{(g_0\eta_{(k)} + g_1\eta_{(k-1)} \cdots g_{n_g}\eta_{(k-n_g)}) \\ &\quad (g_0\eta_{(k-n)} + g_1\eta_{(k-1-n)} \cdots g_{n_g}\eta_{(k-n_g-n)})\} \ . \end{aligned} \tag{12}$$

Considering the ergodicity property (2) of the input white noise, we can write

$$\begin{aligned} R_{\eta\eta}(n) \ &= E\{\eta_{(k)}\eta_{(k-n)}\} \\ &= E\{\eta_{(k+i)}\eta_{(k+i-n)}\} \quad \forall i \in \mathbb{Z} \ . \end{aligned} \tag{13}$$

Then, Eq (12) becomes

$$\begin{aligned} R_{\epsilon\epsilon}(n) \ &= \sum_{i=0}^{n_g}\sum_{j=0}^{n_g} g_i g_j E\{\eta_{(k-i)}\eta_{(k-j-n)}\} \\ &= \sum_{i=0}^{n_g}\sum_{j=0}^{n_g} g_i g_j R_{\eta\eta}(j - i + n) \ . \end{aligned} \tag{14}$$

The autocorrelation function $R_{\eta\eta}(n)$ of the white noise input $\eta_{(k)}$ has the properties of Dirac delta function [16, 19]

$$R_{\eta\eta}(n) = \begin{cases} \sigma_\eta^2 & n = 0 \\ 0 & n \neq 0 \end{cases} \ . \tag{15}$$

Considering Eq (15) and the fact that the argument in autocorrelation function $R_{\eta\eta}(j - i + n)$ is zero if $i = j + n$, the double summation in Eq (14) can be reduced to a single summation as [1, 16]

$$R_{\epsilon\epsilon}(n) = \sigma_\eta^2 \sum_{i=0}^{n_g-n} g_i g_{n+i} \ , \tag{16}$$

where $n = 0\ldots n_g$.

Formula in Eq (16) can be interpreted such that the autocorrelation function $R_{\epsilon\epsilon}(n)$ of the colored noise $\epsilon_{(k)}$, which results from filtering the white noise by a linear system, is equal to the autocorrelation function of its impulse function $g_i$ multiplied by the input noise variance $\sigma_\eta^2$ [21].

**Non-stationary process.** We will repeat the above derivation under the assumption of non-stationary moving average process defined by Eq (6) with time-varying coefficients.

Substituting $\epsilon_{(k)}$ and $\epsilon_{(k+n)}$ in the terms of Eq (7) into Eq (10) yields

$$R_{\epsilon\epsilon}(n,k) = E\left\{ \sum_{i=0}^{n_g} g_i(k)\eta_{(k-i)} \sum_{j=0}^{n_g} g_j(k-n)\eta_{(k-j-n)} \right\} . \tag{17}$$

By expanding the product of summations in Eq (17) we obtain

$$
\begin{aligned}
R_{\epsilon\epsilon}(n,k) \quad &= E\{(g_0(k)\eta_{(k)} + g_1(k)\eta_{(k-1)} \cdots g_{n_g}(k)\eta_{(k-n_g)}) \\
&\quad (g_0(k-n)\eta_{(k-n)} + g_1(k-n)\eta_{(k-1-n)} \cdots g_{n_g}(k-n)\eta_{(k-n_g-n)})\} .
\end{aligned}
\tag{18}
$$

Considering the ergodicity property of the input white noise according to Eqs (2) and (18) becomes

$$
\begin{aligned}
R_{\epsilon\epsilon}(n,k) \quad &= \sum_{i=0}^{n_g}\sum_{j=0}^{n_g} g_i(k)g_j(k-n)E\{\eta_{(k-i)}\eta_{(k-j-n)}\} \\
&= \sum_{i=0}^{n_g}\sum_{j=0}^{n_g} g_i(k)g_j(k-n)R_{\eta\eta}(j-i+n) .
\end{aligned}
\tag{19}
$$

Considering Eq (15) and the fact that the argument in autocorrelation function $R_{\eta\eta}(j-i+n)$ is zero if $i = j + n$, the double summation in Eq (19) can be reduced to a single summation as

$$R_{\epsilon\epsilon}(n,k) = \sigma_\eta^2 \sum_{i=0}^{n_g-n} g_i(k-n)g_{n+i}(k) , \tag{20}$$

where $n = 0\ldots n_g$.

One can notice that the only difference between Eqs (16) and (20) is that the coefficients $g_i(k-n)$, $g_{n+i}(k)$ are sample-dependent in the latter case.

## Exponentially weighted recursive estimate of the autocorrelation function of the colored noise

The autocorrelation function $R_{\epsilon\epsilon}(n)$ needs to be estimated from the available data. Considering a finite-length experiment comprising $N$ samples, the autocorrelation function $R_{\epsilon\epsilon}(n)$ is estimated as the sample autocorrelation function according to formula [19, 20]

$$\hat{R}_{\epsilon\epsilon}(n) = \frac{1}{N-n} \sum_{i=1}^{N-n} \epsilon_{(i)}\epsilon_{(i+n)} , \tag{21}$$

where $n \in \mathbb{N}$ generally satisfies $n \ll N$, yet in our case it will satisfy $n \leq n_g$ due to Eqs (16) and (20).

Now we introduce a novel formula to determine the exponentially weighted sample autocorrelation function as

$$\hat{R}_{\epsilon\epsilon}(n) = \frac{1}{\sum_{i=1}^{N-n} \lambda^{(N-n-i)}} \sum_{i=1}^{N-n} \lambda^{(N-n-i)} \epsilon_{(i)}\epsilon_{(i+n)} , \tag{22}$$

where $0 < \lambda < 1$ is the forgetting factor.

Note that if $\lambda = 1$, then Eq (22) is equivalent to standard formula defined by Eq (21). The rationale for Eq (22) is that it implements the exponential forgetting of older samples where

the latest sample is weighted as $\lambda^{(N-n-N+n)} = \lambda^0 = 1$, whereas the oldest is weighted as $\lambda^{(N-n-1)}$, what implies $\lim_{N \to \infty} \lambda^{(N-n-1)} = 0$ for the limit number of samples.

It is also important to remark that the sample autocorrelation function obtained according to Eqs (21) or (22) is always estimated with a limited accuracy, hence $\hat{R}_{\epsilon\epsilon}(n)$ itself is a random variable [22].

The estimate given by Eq (22) can be proved statistically unbiased for stationary $\epsilon$ by taking the expectancy operator as

$$
\begin{aligned}
E\{\hat{R}_{\epsilon\epsilon}(n)\} \quad &= \frac{1}{\sum_{i=1}^{N-n} \lambda^{(N-n-i)}} E\left\{ \sum_{i=1}^{N-n} \lambda^{(N-n-i)} \epsilon_{(i)} \epsilon_{(i+n)} \right\} \\
&= \frac{1}{\sum_{i=1}^{N-n} \lambda^{(N-n-i)}} \sum_{i=1}^{N-n} \lambda^{(N-n-i)} E\{\epsilon_{(i)} \epsilon_{(i+n)}\} \; .
\end{aligned}
\tag{23}
$$

Since $\epsilon$ is stationary, we can show the unbiasedness

$$
\begin{aligned}
E\{\hat{R}_{\epsilon\epsilon}(n)\} \quad &= E\{\epsilon_{(i)} \epsilon_{(i+n)}\} \frac{1}{\sum_{i=1}^{N-n} \lambda^{(N-n-i)}} \sum_{i=1}^{N-n} \lambda^{(N-n-i)} \\
&= E\{\epsilon_{(i)} \epsilon_{(i+n)}\} \; .
\end{aligned}
\tag{24}
$$

For non-stationary processes, the autocorrelation function given by Eq (10) is $k$ (sample) dependent, hence the the estimate by Eq (22) cannot be claimed unbiased.

An interesting implication is that an unbiased estimate in the case of non-stationary processes would require to consider only the last term of the summation in Eq (22) as $\hat{R}_{\epsilon\epsilon}(n, N) = \epsilon_{(N-n)} \epsilon_{(N)}$, what is obviously impractical because the variance of such an estimate would be very high. Instead, it is preferred to rather use the exponentially weighted sample autocorrelation function according to Eq (22) while a reasonable trade-off between the bias and variance of the estimate must be chosen by properly tuning the forgetting factor $\lambda$.

**Recursive form.** In the case of online estimation of autocorrelation function $R_{\epsilon\epsilon}(n, k)$, we will consider a dataset with an incrementing length $N = k$, so recursive formulation of Eq (22) is mandatory. Summation $\sum_{i=1}^{N-n} \lambda^{(N-n-i)}$ in Eq (22) will be further denoted as

$$
s(n, k) = \sum_{i=1}^{k-n} \lambda^{(k-n-i)} \; .
\tag{25}
$$

The recursive formula for effective updating of summation in Eq (25) can be derived as

$$
\begin{aligned}
s(n, k) \quad &= \sum_{i=1}^{k-n} \lambda^{(k-n-i)} \\
&= \lambda \sum_{i=1}^{k-1-n} \lambda^{(k-1-n-i)} + \lambda^0 \\
&= \lambda s(n, k-1) + 1 \; .
\end{aligned}
\tag{26}
$$

Finally, considering notation from Eq (25), the recursive relation to update the exponentially weighted sample autocorrelation function according to Eq (22) can be derived as

$$
\begin{aligned}
\hat{R}_{\epsilon\epsilon}(n, k) &= \frac{1}{s(n,k)} \sum_{i=1}^{k-n} \lambda^{(k-n-i)} \epsilon_{(i)} \epsilon_{(i+n)} \\
&= \frac{1}{s(n,k)} \left( \lambda \sum_{i=1}^{k-1-n} \left[ \lambda^{(k-1-n-i)} \epsilon_{(i)} \epsilon_{(i+n)} \right] + \epsilon_{(k)} \epsilon_{(k-n)} \right) \\
&= \frac{1}{s(n,k)} \left( \lambda s(n, k-1) \hat{R}_{\epsilon\epsilon}(n, k-1) + \epsilon_{(k)} \epsilon_{(k-n)} \right) ,
\end{aligned}
\tag{27}
$$

where $\hat{R}_{\epsilon\epsilon}(n, k-1)$ is the estimate from the previous sample. It can be concluded that, due to the recursive formulation in Eq (27), calculating the whole summation in Eq (22) by processing the full data sequence can be conveniently avoided. This allows to significantly reduce not only the online computational burden of of the estimation algorithm, but also its memory footprint, since the whole time series does not need to be stored.

The forgetting factor $\lambda$ in Eq (27) is considered the tuning parameter, which should be empirically chosen with a regard to the anticipated statistical properties of the estimated process. Higher values of $\lambda$ (approaching 1) should correspond to close-stationary processes with parameters changing very slowly over time, whereas lower values should be used for the estimation of non-stationary processes, the parameters of which are evolving rather rapidly. It can be claimed that the choice of $\lambda$ represents a trade-off between the adaptation rate and the variance of the parameter estimate. A lower $\lambda$ allows for faster adaptation but induces the undesired phenomenon of forgetting useful information comprised in older samples of the dataset causing rapid fluctuations in the estimated parameters and making it more sensitive to outliers in data. A higher $\lambda$ provides more robust and less volatile parameter estimate, but may result in sluggish response to the actual changes in parameters of moving average process.

## Correlation-based estimation of a stationary process

In the case of stationary process, the analytical form of autocorrelation function in Eq (16) can be written for $n = 0 \ldots n_g$ yielding the following system of $n_g + 1$ nonlinear equations with respect to $n_g + 1$ unknown coefficients $g_i$

$$
\begin{aligned}
\hat{R}_{\epsilon\epsilon}(0) &= \sigma_\eta^2 (g_0^2 + g_1^2 + g_2^2 + \cdots g_{n_g}^2) \\
\hat{R}_{\epsilon\epsilon}(1) &= \sigma_\eta^2 (g_0 g_1 + g_1 g_2 + g_2 g_3 + \cdots g_{n_g-1} g_{n_g}) \\
\hat{R}_{\epsilon\epsilon}(2) &= \sigma_\eta^2 (g_0 g_2 + g_1 g_3 + g_2 g_4 + \cdots g_{n_g-2} g_{n_g}) \\
&\vdots \\
\hat{R}_{\epsilon\epsilon}(n_g - 1) &= \sigma_\eta^2 (g_0 g_{n_g-1} + g_1 g_{n_g}) \\
\hat{R}_{\epsilon\epsilon}(n_g) &= \sigma_\eta^2 g_0 g_{n_g} .
\end{aligned}
\tag{28}
$$

Notice that the true autocorrelation function $R_{\epsilon\epsilon}(n)$ was replaced by the sample-based estimate $\hat{R}_{\epsilon\epsilon}(n)$.

The above equation system can be noted also in the matrix-vector form

$$
\begin{pmatrix}
\hat{R}_{\epsilon\epsilon}(0) \\
\hat{R}_{\epsilon\epsilon}(1) \\
\hat{R}_{\epsilon\epsilon}(2) \\
\vdots \\
\hat{R}_{\epsilon\epsilon}(n_g-1) \\
\hat{R}_{\epsilon\epsilon}(n_g)
\end{pmatrix}
=
\begin{pmatrix}
g_0^2 & g_1^2 & g_2^2 & \cdots & g_{n_g-1}^2 & g_{n_g}^2 \\
g_0 g_1 & g_1 g_2 & g_2 g_3 & \cdots & g_{n_g-1} g_{n_g} & 0 \\
g_0 g_2 & g_1 g_3 & g_2 g_4 & \cdots & 0 & 0 \\
\vdots & \vdots & \vdots & \ddots & \vdots & \vdots \\
g_0 g_{n_g-1} & g_1 g_{n_g} & 0 & \cdots & 0 & 0 \\
g_0 g_{n_g} & 0 & 0 & \cdots & 0 & 0
\end{pmatrix}
\begin{pmatrix}
1 \\
1 \\
1 \\
\vdots \\
1 \\
1
\end{pmatrix}
\sigma_\eta^2 .
\tag{29}
$$

We will further use the shorthand notation for Eq (29) as

$$
\hat{\mathbf{R}}_{\epsilon\epsilon} = \mathbf{c}(\mathbf{g})\sigma_\eta^2 ,
\tag{30}
$$

where $\mathbf{c}(\mathbf{g}) : \mathbb{R}^{n_g+1} \to \mathbb{R}^{n_g+1}$ is a nonlinear multivariate vector function and vector $\mathbf{g}$ is defined by Eq (5).

The goal of the identification is to find a solution $\mathbf{g}^*$ of nonlinear equation system in Eq (29) satisfying $\hat{\mathbf{R}}_{\epsilon\epsilon} - \mathbf{c}(\mathbf{g}^*)\sigma_\eta^2 = \mathbf{0}$.

**Existence of a solution.** Equation system given by Eq (29) comprises one equation for a hypersphere (the first equation) and $n_g$ equations for hyperbolas (the remaining equations), while each of the hyperbolas is dependent on a different number of variables. The constant left-hand term, which is equal to the sample autocorrelation function $\hat{R}_{\epsilon\epsilon}(i)$, determines the squared radius of a hypersphere or the curvature of hyperbolas. A curve defined by the intersections of $n_g$ hyperbolas in the $n_g + 1$ dimensional space can intersect the hypersphere either in 4, 2 or zero points.

Considering the solution $\mathbf{g}^*$ of nonlinear equation system in Eq (29), which satisfies $\hat{\mathbf{R}}_{\epsilon\epsilon} - \mathbf{c}(\mathbf{g}^*)\sigma_\eta^2 = \mathbf{0}$, there can be directly determined also the complementary solution $-\mathbf{g}^*$ satisfying $\hat{\mathbf{R}}_{\epsilon\epsilon} - \mathbf{c}(-\mathbf{g}^*)\sigma_\eta^2 = \mathbf{0}$, since all the terms in Eq (29) are either bilinear or quadratic, what implies that $(-g_i)^2 = g_i^2$ and $g_i g_j = (-g_i)(-g_j)$. Moreover, looking at the system of equations in Eq (29), one can find out that its solution $\mathbf{g}^*$ is symmetrical with respect to the order of coefficients. Therefore, having the original solution $\mathbf{g}^*$, another vector $\mathbf{g}^\dagger$ also satisfies $\hat{\mathbf{R}}_{\epsilon\epsilon} - \mathbf{c}(\mathbf{g}^\dagger)\sigma_\eta^2 = \mathbf{0}$ if defined as

$$
\mathbf{g}^\dagger = \mathcal{F}\mathbf{g}^* ,
\tag{31}
$$

where $\mathcal{F}$ is the exchange matrix i.e. the backward identity (anti-diagonal) matrix.

Finally, we have four solutions of the equation system given by Eq (29) defined as

$$
\mathbf{g}^*, \ -\mathbf{g}^*, \ \mathcal{F}\mathbf{g}^*, \ -\mathcal{F}\mathbf{g}^* .
\tag{32}
$$

The above result can be interpreted such that changing the order or the sign of all coefficients do not affect the properties of the moving average process in the correlation domain.

Concerning the solvability of equation system given by Eq (29), i.e. the existence of all four solutions in Eq (32), the following important property directly results from Eq (29).

First, we can write

$$
\begin{aligned}
(g_0 - g_1 + g_2 \ldots \pm g_{n_g-1} \mp g_{n_g})^2 = \quad & (g_0^2 + g_1^2 + g_2^2 \ldots + g_{n_g-1}^2 + g_{n_g}^2) \\
& -2(g_0g_1 + g_1g_2 + g_2g_3 + \ldots + g_{n_g-1}g_{n_g}) \\
& +2(g_0g_2 + g_1g_3 + g_2g_4 + \ldots + g_{n_g-2}g_{n_g}) \\
& -2(g_0g_3 + g_1g_4 + g_2g_5 + \ldots + g_{n_g-3}g_{n_g}) \qquad (33) \\
& \quad \vdots \\
& \pm 2(g_0g_{n_g-1} + g_1g_{n_g}) \\
& \mp 2g_0g_{n_g} \; .
\end{aligned}
$$

Realizing that the quadratic term $(g_0 - g_1 + g_2 \ldots \pm g_{n_g-1} \mp g_{n_g})^2 = \rho^2$ represents a non-negative quantity, we can state the following inequality noted in the matrix form

$$
\mathbf{1}^{\mathrm{T}}
\begin{pmatrix}
2g_0g_1 & 2g_1g_2 & 2g_2g_3 & \ldots & 2g_{n_g-1}g_{n_g} & 0 \\
-2g_0g_2 & -2g_1g_3 & -2g_2g_4 & \ldots & 0 & 0 \\
\vdots & \vdots & \vdots & \ddots & \vdots & \vdots \\
\mp 2g_0g_{n_g-1} & \mp 2g_1g_{n_g} & 0 & \ldots & 0 & 0 \\
\pm 2g_0g_{n_g} & 0 & 0 & \ldots & 0 & 0
\end{pmatrix}
\mathbf{1} \leq g_0^2 + g_1^2 + g_2^2 \ldots + g_{n_g-1}^2 + g_{n_g}^2 \;, (34)
$$

where $\mathbf{1} \in \mathbb{R}^{n_g+1}$ is the vector of ones. Considering the system of equations in (29), the above inequality results in the necessary condition for the existence of a solution as

$$
2\left( \hat{R}_{\epsilon\epsilon}(1) - \hat{R}_{\epsilon\epsilon}(2) + \ldots \pm \hat{R}_{\epsilon\epsilon}(n_g - 1) \mp \hat{R}_{\epsilon\epsilon}(n_g) \right) \leq \hat{R}_{\epsilon\epsilon}(0) \; . \tag{35}
$$

**Numerical solution.** It is apparent that the equation system in Eq (29) cannot be solved analytically. Therefore, to find the solution by the numeric means, the error function $\mathbf{e}(\mathbf{g}) : \mathbb{R}^{n_g+1} \to \mathbb{R}^{n_g+1}$ will be introduced as

$$
\mathbf{e}(\mathbf{g}) = \mathbf{c}(\mathbf{g})\sigma_\eta^2 - \hat{\mathbf{R}}_{\epsilon\epsilon} \;, \tag{36}
$$

where the individual elements of this multivariate vector function $\mathbf{e}(\mathbf{g})$ are defined as

$$
\mathbf{e}(\mathbf{g}) = \begin{bmatrix} e_0(\mathbf{g}) & e_1(\mathbf{g}) & e_2(\mathbf{g}) & \ldots & e_{n_g}(\mathbf{g}) \end{bmatrix}^{\mathrm{T}} \;. \tag{37}
$$

The considered problem is to find a root of the error function defined by Eq (36), i.e. a solution vector satisfying $\mathbf{e}(\mathbf{g}^\star) = \mathbf{0}$. To this end, we will harness iterative numeric algorithms suitable for solving nonlinear multivariate problems, in particular, the Newton-Raphson and the Levenberg–Marquardt algorithm.

The iteration of Newton-Raphson algorithm gets [23, 24]

$$
\mathbf{g}_{p+1} = \mathbf{g}_p - [J(\mathbf{g}_p)]^{-1}\mathbf{e}(\mathbf{g}_p) \;, \tag{38}
$$

where $p$ is the iteration number and $J(\mathbf{g}) : \mathbb{R}^{n_g+1} \to \mathbb{R}^{n_g+1 \times n_g+1}$ is the Jacobian matrix of function $\mathbf{e}(\mathbf{g})$ with respect to vector $\mathbf{g}$.

The Jacobian matrix of the error function given by Eq (36) can be derived according to Eq (28) as

$$
J(\mathbf{g}) = \sigma_\eta^2
\begin{pmatrix}
2g_0 & 2g_1 & 2g_2 & \cdots & 2g_{n_g-1} & 2g_{n_g} \\
g_1 & g_2 + g_0 & g_3 + g_1 & \cdots & g_{n_g} + g_{n_g-2} & g_{n_g-1} \\
g_2 & g_3 & g_4 + g_0 & \cdots & g_{n_g-1} & g_{n_g-2} \\
\vdots & \vdots & \vdots & \ddots & \vdots & \vdots \\
g_{n_g-1} & g_{n_g} & 0 & \cdots & g_0 & g_1 \\
g_{n_g} & 0 & 0 & \cdots & 0 & g_0
\end{pmatrix} .
\tag{39}
$$

Alternatively, to reduce the sensitivity with respect to the choice of the initial guess $\mathbf{g}_0$, one may use the Levenberg–Marquardt algorithm with the added damping term [23, 24] as

$$
\mathbf{g}_{p+1} = \mathbf{g}_p - [J(\mathbf{g}_p) + \kappa_p \ \mathrm{diag}(J(\mathbf{g}_p))]^{-1} \mathbf{e}(\mathbf{g}_p) ,
\tag{40}
$$

where $\kappa_p$ is the iteration-dependent damping coefficient that follows the heuristic rule

$$
\kappa_p = \frac{\kappa_0}{v^p} \quad 10^1 < \kappa_0 < 10^5 \quad v > 1 .
\tag{41}
$$

The convergence of numeric solution will be quantified by the scalar criterion representing the sum of the squared errors

$$
\mathcal{J}(p) = \mathbf{e}(\mathbf{g}_p)^{\mathrm{T}} \mathbf{e}(\mathbf{g}_p) .
\tag{42}
$$

A drop in this criterion below a certain threshold can also be used to trigger the termination of the Newton-Raphson and Levenberg-Marquardt algorithms.

It is worth mentioning that an emerging issue with algorithms defined by Eqs (38) and (40) is the need to determine the Jacobian matrix inverse $[J(\mathbf{g}_p)]^{-1}$ or the regularized Jacobian matrix inverse $[J(\mathbf{g}_p) + \kappa_p \ \mathrm{diag} \ (J(\mathbf{g}_p))]^{-1}$ at each algorithm iteration $p$. Considering a nonzero update of all the elements of the parameter vector as $\mathbf{g}_{p+1} = \mathbf{g}_p + \Delta\mathbf{g}_p$ in the terms of iteration given by Eqs (38) or (40), the Jacobian defined by Eq (39), which is linear with respect to the coefficients $g_i$, will be updated linearly as

$$
J(\mathbf{g}_{p+1}) = J(\mathbf{g}_p + \Delta\mathbf{g}_p) = J(\mathbf{g}_p) + J(\Delta\mathbf{g}_p) .
\tag{43}
$$

Eq (43) implies that the Jacobian perturbation $J(\Delta\mathbf{g}_p)$ is a full-rank matrix because the Jacobian itself must have the full rank to be invertible. It means that the perturbation $J(\Delta\mathbf{g}_p)$ of the Jacobian matrix given by Eq (39) caused by the update of the estimated parameters cannot be written as an outer product of vectors i.e. as $\mathbf{u}\mathbf{v}^T$. As a consequence, the well-known Sherman–Morrison formula or the Woodbury matrix identity [25, 26] cannot be used to effectively update the perturbed Jacobian inverse.

Fortunately, one can find out that there is no need to perform the explicit matrix inverse at all since evaluating the term $[J(\mathbf{g}_p)]^{-1} \mathbf{e}(\mathbf{g}_p)$ in the Newton-Raphson iteration given by Eq (38) and evaluating term $[J(\mathbf{g}_p) + \kappa_p \ \mathrm{diag}(J(\mathbf{g}_p))]^{-1} \mathbf{e}(\mathbf{g}_p)$ in the Levenberg–Marquardt iteration given by Eq (40) can be equivalently interpreted as finding solutions to the linear equation system $J(\mathbf{g}_p)(\mathbf{g}_p - \mathbf{g}_{p+1}) = \mathbf{e}(\mathbf{g}_p)$ or $[J(\mathbf{g}_p) + \kappa_p \ \mathrm{diag}(J(\mathbf{g}_p))](\mathbf{g}_p - \mathbf{g}_{p+1}) = \mathbf{e}(\mathbf{g}_p)$ respectively, which

can be obtained effectively by LU decomposition [27] with $\mathcal{O}((n_g + 1)^3 + 2(n_g + 1)^2) = \mathcal{O}((n_g + 1)^3)$ complexity [26, 28].

If the equation system given by Eq (30) has a solution, the numerical approximation of the solution will converge to one of four solutions in Eq (32) depending on the choice of the initial guess $\mathbf{g}_0$. It can be showed that the choice of the initial guess $\mathbf{g}_0$ affects the convergence as follows:

$$
\begin{aligned}
\mathbf{g}_0 &= \begin{bmatrix} 1 & 0 & 0 & \ldots & 0 \end{bmatrix}^{\mathrm{T}} \rightarrow \mathbf{g}^* \quad \text{original sign and order}, \\
\mathbf{g}_0 &= -\begin{bmatrix} 1 & 0 & 0 & \ldots & 0 \end{bmatrix}^{\mathrm{T}} \rightarrow -\mathbf{g}^* \quad \text{different sign}, \\
\mathbf{g}_0 &= \begin{bmatrix} 0 & 0 & 0 & \ldots & 1 \end{bmatrix}^{\mathrm{T}} \rightarrow \mathcal{L}\mathbf{g}^* \quad \text{different order}, \\
\mathbf{g}_0 &= -\begin{bmatrix} 0 & 0 & 0 & \ldots & 1 \end{bmatrix}^{\mathrm{T}} \rightarrow -\mathcal{L}\mathbf{g}^* \quad \text{different order and sign},
\end{aligned}
\tag{44}
$$

which implies that the initial guess $\mathbf{g}_0$ should be chosen to represent the white noise model $\epsilon_{(k)} = \eta_{(k)}$. Note that each of the four solutions in Eq (32) can be easily transformed to the desired form by modifying the sign or the order of the coefficients.

The block diagram of the proposed algorithm for estimation of stationary moving average processes is depicted in Fig 1.

## Correlation-based estimation of a non-stationary process

In the case of non-stationary process, the analytical form of the autocorrelation function given by Eq (20) can be written for $n = 0 \ldots n_g$ yielding the following system of $n_g$ linear equations and one nonlinear equation with respect to $n_g + 1$ unknown coefficients $g_i(k)$

$$
\begin{aligned}
\hat{R}_{\epsilon\epsilon}(0, k) &= \sigma_\eta^2 \Big( g_0^2(k) + g_1^2(k) + g_2^2(k) + \ldots g_{n_g}^2(k) \Big) \\[4pt]
\hat{R}_{\epsilon\epsilon}(1, k) &= \sigma_\eta^2 (g_0(k-1)g_1(k) + g_1(k-1)g_2(k) + g_2(k-1)g_3(k) + \ldots g_{n_g-1}(k-1)g_{n_g}(k)) \\[4pt]
\hat{R}_{\epsilon\epsilon}(2, k) &= \sigma_\eta^2 (g_0(k-2)g_2(k) + g_1(k-2)g_3(k) + g_2(k-2)g_4(k) + \ldots g_{n_g-2}(k-2)g_{n_g}(k)) \\[4pt]
&\quad \vdots \\[4pt]
\hat{R}_{\epsilon\epsilon}(n_g - 1, k) &= \sigma_\eta^2 (g_0(k - n_g + 1)g_{n_g-1}(k) + g_1(k - n_g + 1)g_{n_g}(k)) \\[4pt]
\hat{R}_{\epsilon\epsilon}(n_g, k) &= \sigma_\eta^2 g_0(k - n_g)g_{n_g}(k),
\end{aligned}
\tag{45}
$$

where $g_i(k-1) \ldots g_i(k - n_g)$ represent the estimates of coefficients obtained from the previous samples (iterations).

The above equation system can be split into one nonlinear equation

$$
\hat{R}_{\epsilon\epsilon}(0, k) = \sigma_\eta^2 \Big( g_0^2(k) + g_1^2(k) + g_2^2(k) + \ldots g_{n_g}^2(k) \Big),
\tag{46}
$$

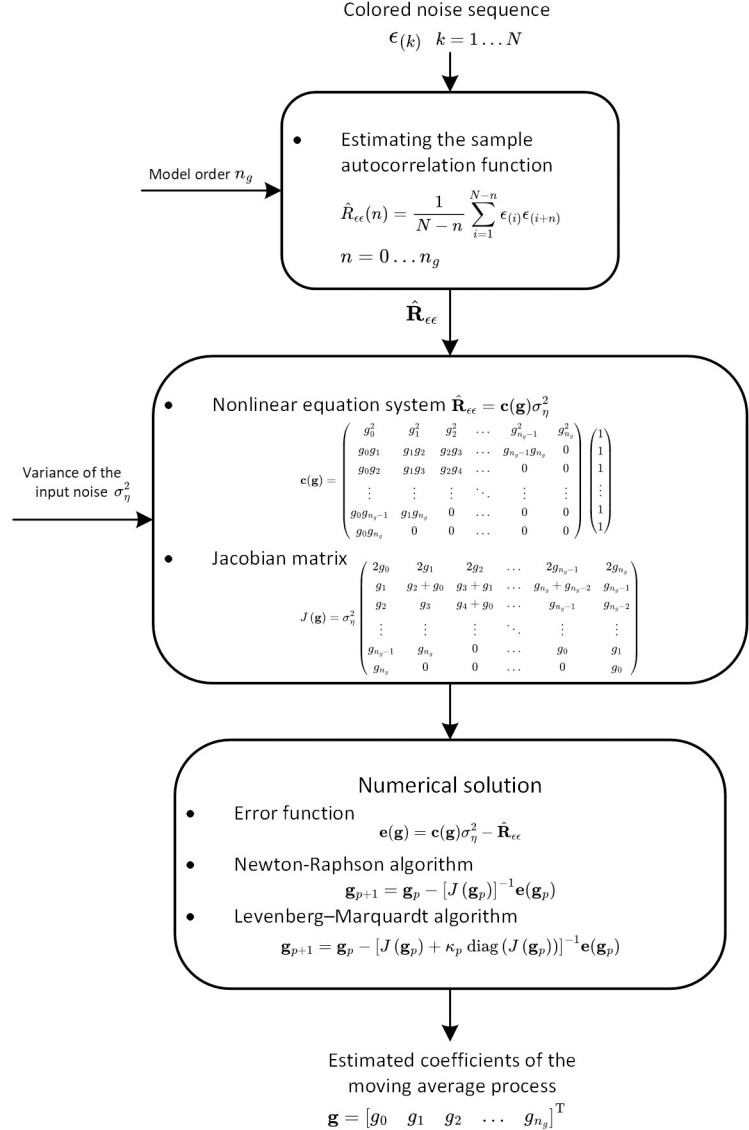

**Fig 1. Block diagram of the proposed algorithm for estimation of stationary moving average processes.**

and a system of $n_g$ linear equations noted as

$$
\begin{pmatrix}
\hat{R}_{\epsilon\epsilon}(1,k) \\
\hat{R}_{\epsilon\epsilon}(2,k) \\
\vdots \\
\hat{R}_{\epsilon\epsilon}(n_g-1,k) \\
\hat{R}_{\epsilon\epsilon}(n_g,k)
\end{pmatrix}
=
\begin{pmatrix}
g_0(k-1) & g_1(k-1) & \cdots & g_{n_g-2}(k-1) & g_{n_g-1}(k-1) \\
0 & g_0(k-2) & \cdots & g_{n_g-3}(k-2) & g_{n_g-2}(k-2) \\
\vdots & \vdots & \ddots & \vdots & \vdots \\
0 & 0 & \cdots & g_0(k-n_g+1) & g_1(k-n_g+1) \\
0 & 0 & \cdots & 0 & g_0(k-n_g)
\end{pmatrix}
\begin{pmatrix}
g_1(k) \\
g_2(k) \\
\vdots \\
g_{n_g-1}(k) \\
g_{n_g}(k)
\end{pmatrix}
\sigma_\eta^2 \,. (47)
$$

The linear equation system in Eq (47) can be interpreted such that the solution of the estimation problem to obtain the current estimate $\mathbf{g}(k)$ is linearly dependent on the previous

estimates $\mathbf{g}(k-1)\ldots\mathbf{g}(k-n_g)$. Therefore, the estimation algorithm is recursive not only because we update the sample autocorrelation function $\hat{R}_{\epsilon\epsilon}(n,k)$ recursively according to Eq (27), but also because we determine the parameter estimate $\mathbf{g}(k)$ based on the estimates from previous iterations.

Another significant advantage is that the matrix of linear equation system in Eq (47) is triangular, hence it can be solved directly by the backward substitution according to Eq (48) without a need for performing the LU decomposition first, which reduces the computational complexity from $\mathcal{O}(n_g^3)$ to $\mathcal{O}(n_g^2)$.

$$g_i(k) = \frac{\hat{R}_{\epsilon\epsilon}(i,k) - \sum_{j=1}^{n_g-i} g_j(k-i)g_{i+j}(k)}{\sigma_\eta^2 g_0(k-i)} \quad \forall i = n_g, n_g - 1 \ldots 1 \tag{48}$$

Finally, according to Eq (45) the remaining coefficient $g_0(k)$ can be obtained by solving the first nonlinear equation as

$$g_0(k) = \sqrt{\frac{\hat{R}_{\epsilon\epsilon}(0,k)}{\sigma_\eta^2} - \sum_{i=1}^{n_g} g_i^2(k)} \ . \tag{49}$$

The block diagram of the proposed algorithm for online estimation of non-stationary moving average processes is depicted in Fig 2.

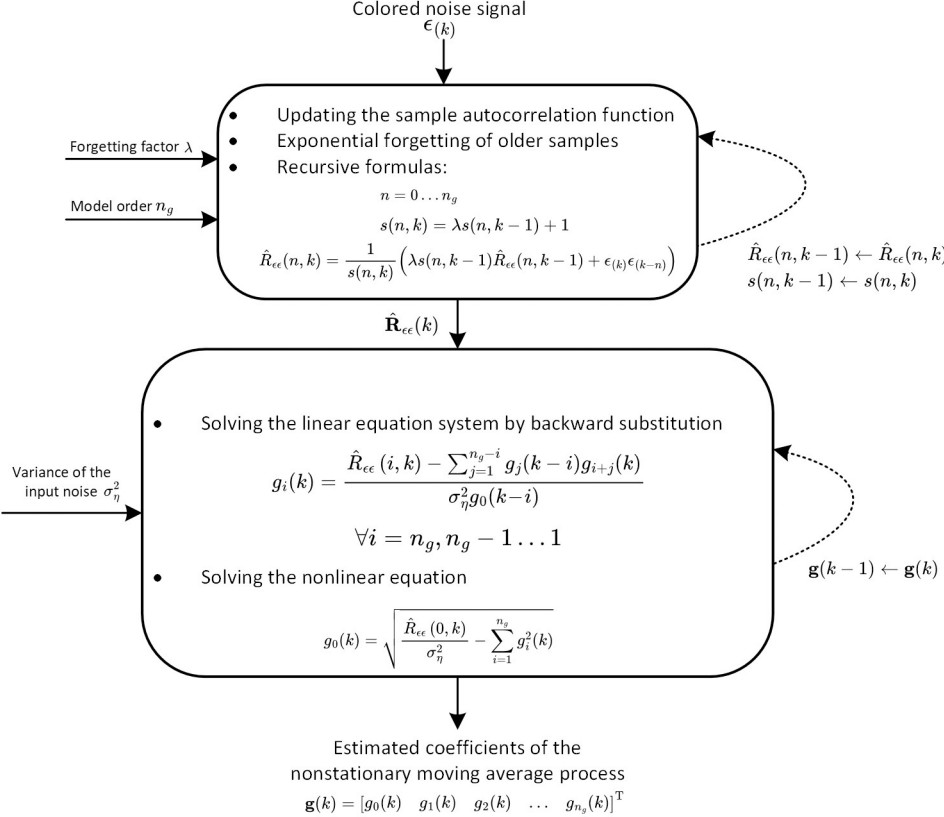

**Fig 2. Block diagram of the proposed algorithm for online estimation of non-stationary moving average processes.**

**Computational complexity and memory footprint analysis.** Now analyze the overall memory footprint and per-iteration computational complexity of the proposed correlation-based method for online estimation of non-stationary process.

Based on the linear equation system given by Eq (47), it can be concluded that there needs to be stored maximally $n_g$ model coefficients estimated from $n_g$ previous algorithm iterations, which implies $\frac{n_g(n_g+1)}{2}$ variables. Moreover, the sample autocorrelation function $\hat{R}_{\epsilon\epsilon}(n,k)$ and the values of individual summations $s(n,k)$ need to be stored for $n = 0 \ldots n_g$, which results in additional $2(n_g + 1)$ real-valued variables. To recursively update the sample autocorrelation function according to Eq (27) for $n = 0 \ldots n_g$, there needs to be stored $n_g + 1$ past values of the colored noise $\epsilon_{(k-n)}$. Summarizing the above gives the minimal memory footprint of $\frac{n_g(n_g+1)}{2} + 3(n_g + 1)$ real-valued variables.

Concerning the computational complexity per algorithm iteration, recursively updating the individual summations $s(n,k)$ for $n = 0 \ldots n_g$ according to (26) represents a linear complexity $n_g + 1$, and the recursive update of the sample autocorrelation function $\hat{R}_{\epsilon\epsilon}(n,k)$ for arguments $n = 0 \ldots n_g$ according to Eq (27) also has a linear complexity $n_g + 1$. Solving the triangular system of linear equations given by Eq (47) according to Eq (48) can be achieved with a reduced quadratic complexity of $\frac{n_g(n_g-1)}{2}$. Finally, the last computation given by Eq (49) can be achieved with linear complexity $n_g$. Considering big O notation, the asymptotic per-iteration computational complexity is $\mathcal{O}(n_g^2)$.

## Results and discussion

In this section, we will evaluate and discuss multiple experiments to validate the proposed methodology. The validation will be carried out considering offline scenario for estimating stationary process and online scenario for estimating non-stationary process.

### Stationary processes

Consider the following stationary moving average process

$$\epsilon_{(k)} = (1.5 + 0.9z^{-1} + 0.5z^{-2} + 0.4z^{-3} + 0.25z^{-4} + 0.2z^{-5})\eta_{(k)} \; , \tag{50}$$

which is a representative of the generic structure defined by Eq (1).

The sequence of colored noise signal $\epsilon$ was obtained by filtering a randomly generated white noise sequence with the variance $\sigma_\eta^2 = 3.5$ and length $N = 5000$ samples by the moving average model given by Eq (50). The corresponding timeseries of $\eta$ is in S1 Dataset.

The actual autocorrelation function $R_{\epsilon\epsilon}(n)$ determined according to Eq (16) and the sample autocorrelation function $\hat{R}_{\epsilon\epsilon}(n)$ estimated offline according to Eq (21) are plotted in Fig 3, where a relatively small estimate error can be observed.

Notice that $\hat{R}_{\epsilon\epsilon}(n)$ can be considered virtually zero for the lag arguments $n$ larger than five $(n_g)$, what suggests the model order $n_g = 5$. Such a visual analysis of the sample autocorrelation function can be eventually used to determine the correct model order $n_g$ if it is considered unknown.

Analyzing the necessary condition for solvability of the estimation problem based on the sample autocorrelation function from Fig 3, the inequality condition given by Eq (35) is satisfied as $12.699 > 10.9318$, hence the solution of the autocorrelation matching problem can be found.

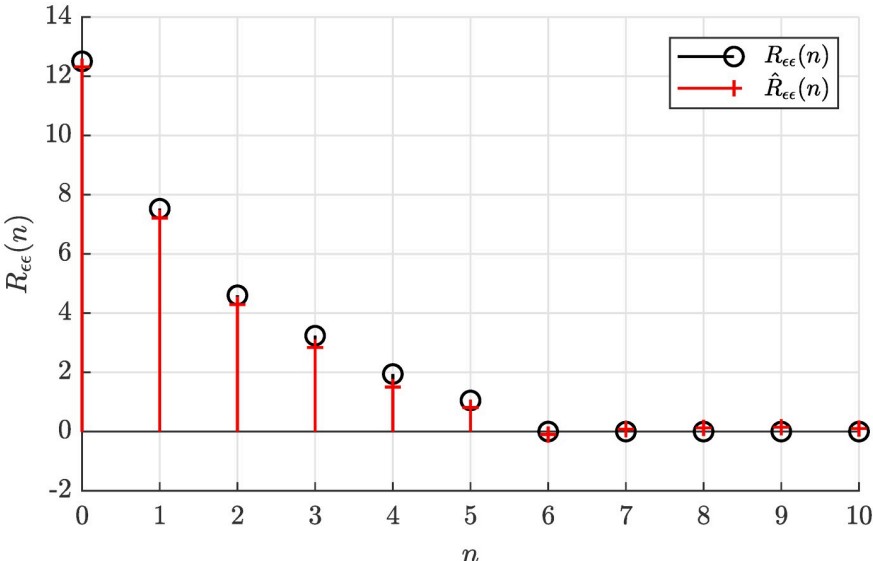

**Fig 3. Actual autocorrelation function $R_{\epsilon\epsilon}(n)$ and the estimated autocorrelation function $\hat{R}_{\epsilon\epsilon}(n)$ for $N = 5000$.**

By performing the offline identification while considering the pseudolinear regression strategy, the method of Durbin and finally the proposed correlation method respectively, the following coefficients were estimated.

Using the pseudolinear regression strategy with applied 10 iterations:

$$\mathbf{g} = \begin{bmatrix} 1.5726 & 0.8528 & 0.3951 & 0.2540 & 0.0549 & 0.0238 \end{bmatrix}^{\mathrm{T}} \tag{51}$$

Using the method of Durbin:

$$\mathbf{g} = \begin{bmatrix} 1.5199 & 0.8854 & 0.4874 & 0.3714 & 0.1678 & 0.1765 \end{bmatrix}^{\mathrm{T}} \tag{52}$$

Using the proposed autocorrelation matching method by applying 15 iterations of the Newton-Raphson algorithm:

$$\mathbf{g} = \begin{bmatrix} 1.5145 & 0.8860 & 0.4902 & 0.3728 & 0.1938 & 0.1528 \end{bmatrix}^{\mathrm{T}} \tag{53}$$

The estimated and actual coefficients are also visualized in Fig 4 demonstrating that the correlation method and the Durbin's method provided very accurate estimates, while the pseudolinear regression performed slightly worse.

Concerning the convergence of the numerical solution of the estimation problem, we applied both the Newton-Raphson algorithm and the Levenberg–Marquardt algorithm with different parameterization. In Fig 5 one can see that the Levenberg–Marquardt algorithm provided slightly slower convergence of quadratic metric given by Eq (42) compared to the Newton-Raphson algorithm. From Fig 5 it can also be concluded that convergence properties of the Levenberg–Marquardt algorithm can be effectively tuned by choosing the damping factor parameters $\kappa_0$ and $\nu$.

By filtering the output colored noise signal $\epsilon$ by the inverse $\frac{1}{g(z)}$ of the identified moving average model applying Eq (8), the input noise sequence $\hat{\eta}$ was determined. The sample autocorrelation function $\hat{R}_{\hat{\eta}\hat{\eta}}(n)$ of the estimated input noise sequence is depicted in Fig 6. This

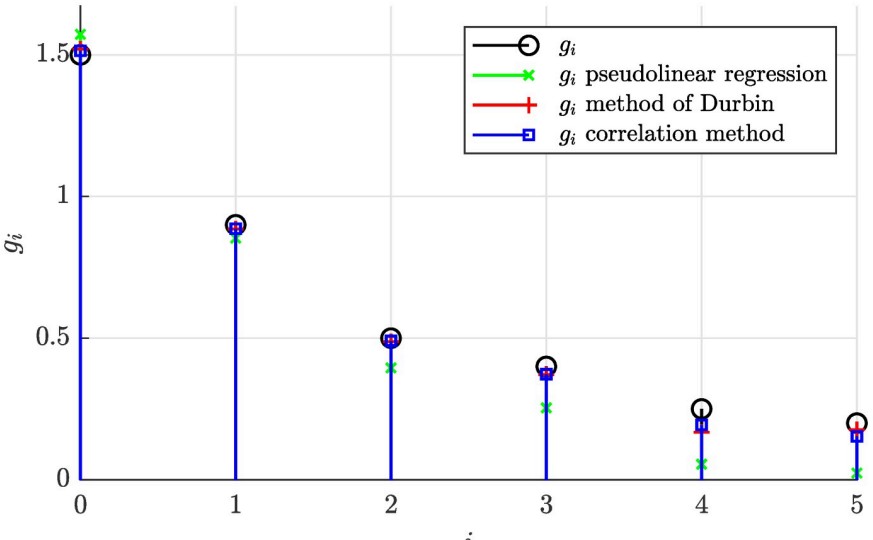

**Fig 4. Actual and estimated coefficients of the stationary moving average process for *N* = 5000.**

autocorrelation function shows the Dirac delta-like character what proves the identified models valid from the autocorrelation perspective.

Now lower the number of available samples of the colored noise signal to $N = 500$ and then estimate the parameters of the process. The corresponding timeseries of $\eta$ is in S2 Dataset.

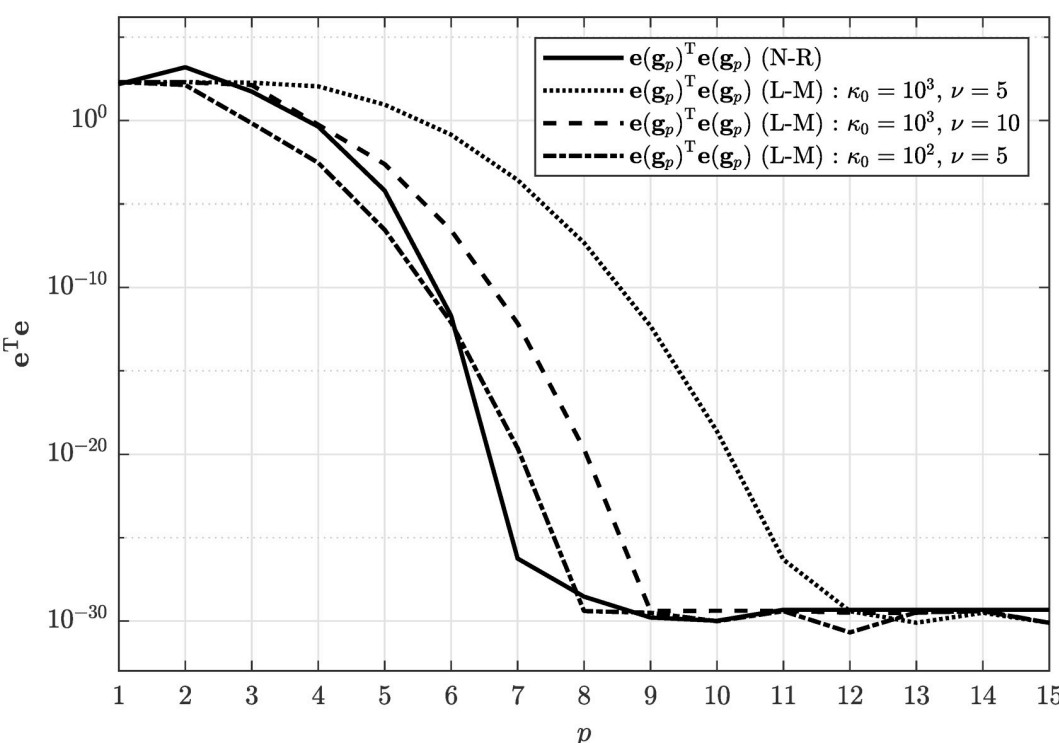

**Fig 5. Convergence of the numerical solution of the non-linear equation system through the iterations *p* using the Newton-Raphson algorithm and the Levenberg–Marquardt algorithm.**

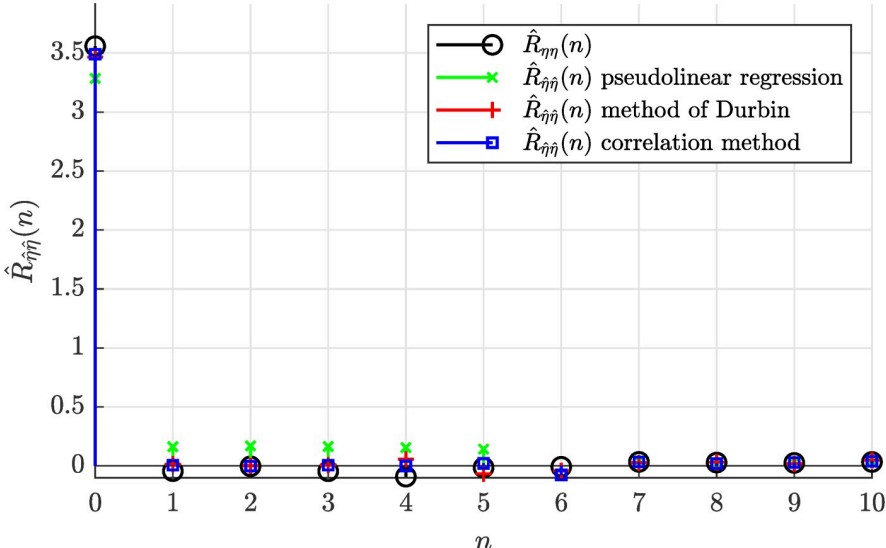

**Fig 6. Estimate of the autocorrelation function $\hat{R}_{\hat{\eta}\hat{\eta}}(n)$ of the estimated noise input sequence for $N = 5000$.**

Fig 7 shows the corresponding sample autocorrelation function, which has a deteriorated accuracy. Evaluating all three considered estimation methods, Fig 8 documents the estimated coefficients obtained from this reduced dataset, which are apparently not as close to the actual values as those in Fig 4.

In order to quantify the performance of the estimation methods, a quadratic estimation accuracy metric $(\mathbf{g}^* - \mathbf{g})^{\mathrm{T}} (\mathbf{g}^* - \mathbf{g})$ was evaluated and the results are summarized in Table 1 for all the considered methods and the number of samples $N$. This table documents that the estimation accuracy turned out to be slightly better in the case of proposed autocorrelation

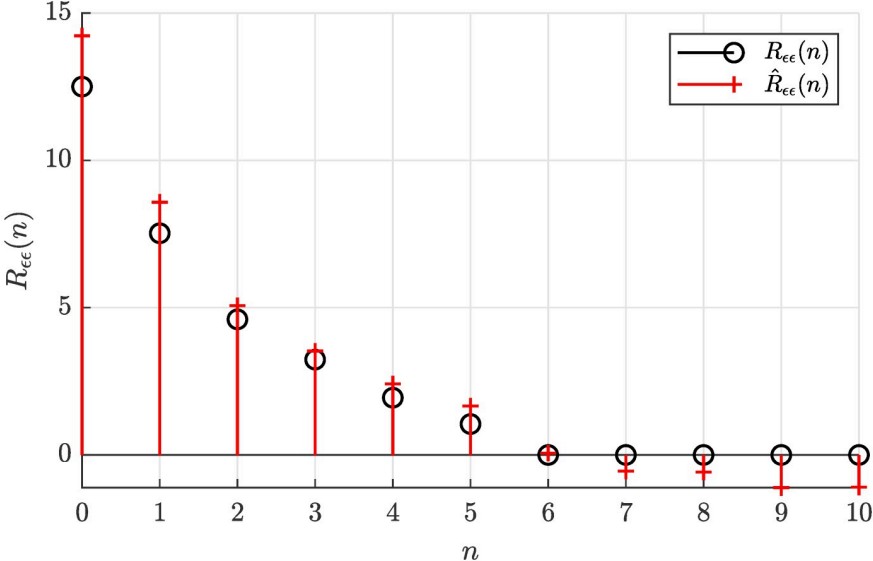

**Fig 7. Actual autocorrelation function $R_{\epsilon\epsilon}(n)$ and the estimated autocorrelation function $\hat{R}_{\epsilon\epsilon}(n)$ for $N = 500$.**

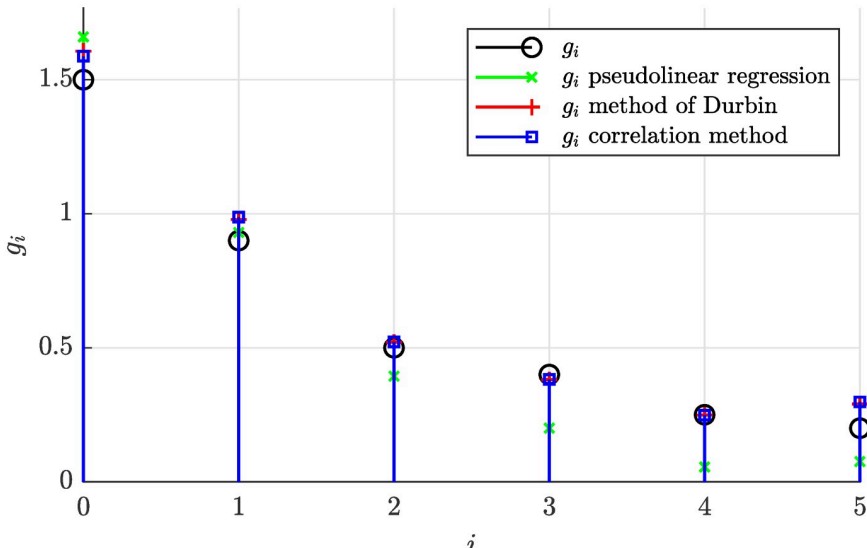

**Fig 8. Actual and estimated coefficients of the stationary moving average process for $N = 500$.**

matching method compared to the method of Durbin, while the pseudolinear regression performed significantly worse.

The presented experiments, which involve data in the form of randomly generated noise sequences of length $N = 5000$ samples, were repeated for multiple ($M$) different realizations of the random white noise sequences $\eta$. The corresponding timeseries of $\eta$ are in S3 Dataset. These realizations yielded the corresponding correlated noise sequences $\epsilon$, which were used for model estimation. For the $i$-th of the total $M = 100$ realizations, the parameter estimate vector $\mathbf{g}_i^*$ was determined using each of the three compared methods, and the corresponding statistics were then evaluated. To assess the performance in the statistical framework, the mean values and the sample variances of the residuals of the estimated parameters were calculated.

The mean value vector of the residuals vector $\mathbf{g}_i^* - \mathbf{g}$ was obtained as

$$\mu_{(\mathbf{g}^*\mathbf{g})} = \frac{1}{M} \sum_{i=1}^{M} \left( \mathbf{g}_i^* - \mathbf{g} \right) . \tag{54}$$

The vector of variances of the residuals $\mathbf{g}_i^* - \mathbf{g}$ was obtained as

$$S_{(\mathbf{g}^*\mathbf{g})}^2 = \frac{1}{M} \sum_{i=1}^{M} \mathrm{diag}((\mathbf{g}_i^* - \mathbf{g})^{\mathrm{T}} (\mathbf{g}_i^* - \mathbf{g})) . \tag{55}$$

The motivation for conducting the statistical analysis described above is to rigorously assess the performance of the compared methods across multiple random realizations of the input

**Table 1. Evaluated estimation accuracy metric for all the considered methods and the number of samples $N$.**

|  | $(\mathbf{g}^* - \mathbf{g})^{\mathrm{T}} (\mathbf{g}^* - \mathbf{g})$, $N = 5000$ | $(\mathbf{g}^* - \mathbf{g})^{\mathrm{T}} (\mathbf{g}^* - \mathbf{g})$, $N = 500$ |
|---|---|---|
| Correlation method | $6.64 \times 10^{-3}$ | $2.58 \times 10^{-2}$ |
| Durbin's method | $8.91 \times 10^{-3}$ | $2.65 \times 10^{-2}$ |
| Pseudolinear regression | $1.08 \times 10^{-1}$ | $1.31 \times 10^{-1}$ |

**Table 2. Statistical evaluation of the estimation accuracy of the stationary moving average process by the sample mean $\mu_{(\mathbf{g}^*-\mathbf{g})}$ and sample variance $S^2_{(\mathbf{g}^*-\mathbf{g})}$ of the estimate residuals $\mathbf{g}^*_i - \mathbf{g}$ for $M = 100$ experiment realizations and $N = 5000$ samples.**

| | $\mu_{(\mathbf{g}^*-\mathbf{g})}$ |
|---|---|
| Correlation method | $(-2.8 \times 10^{-3} \ -5.1 \times 10^{-3} \ -3.2 \times 10^{-3} \ 1.5 \times 10^{-3} \ -1.8 \times 10^{-3} \ -2.9 \times 10^{-3})^{\mathrm{T}}$ |
| Durbin's method | $(5.8 \times 10^{-3} \ -8.1 \times 10^{-3} \ -6.7 \times 10^{-3} \ -2.8 \times 10^{-3} \ -28.2 \times 10^{-3} \ -0.4 \times 10^{-3})^{\mathrm{T}}$ |
| Pseudolinear regression | $(61.6 \times 10^{-3} \ -52.1 \times 10^{-3} \ -117.5 \times 10^{-3} \ -146.3 \times 10^{-3} \ -177.4 \times 10^{-3} \ -190.6 \times 10^{-3})^{\mathrm{T}}$ |
| | $S^2_{(\mathbf{g}^*-\mathbf{g})}$ |
| Correlation method | $(0.2 \times 10^{-3} \ 0.7 \times 10^{-3} \ 0.6 \times 10^{-3} \ 0.8 \times 10^{-3} \ 0.8 \times 10^{-3} \ 2.6 \times 10^{-3})^{\mathrm{T}}$ |
| Durbin's method | $(0.2 \times 10^{-3} \ 0.7 \times 10^{-3} \ 0.6 \times 10^{-3} \ 0.7 \times 10^{-3} \ 1.6 \times 10^{-3} \ 0.8 \times 10^{-3})^{\mathrm{T}}$ |
| Pseudolinear regression | $(4.2 \times 10^{-3} \ 3.3 \times 10^{-3} \ 14.3 \times 10^{-3} \ 22.0 \times 10^{-3} \ 32.1 \times 10^{-3} \ 37.2 \times 10^{-3})^{\mathrm{T}}$ |

noise sequences, thereby canceling the random effects and supporting the validity of the results. The results for the stationary process (50) summarized in Table 2 show that the proposed correlation method provided mean values of the residuals $\mu_{(\mathbf{g}^*-\mathbf{g})}$ closest to zero, while the corresponding variances of the residuals $S^2_{(\mathbf{g}^*-\mathbf{g})}$ were similar to those obtained using Durbin's method. Thus, it can be claimed that their performances are very similar in terms of statistical bias and efficiency. In contrast, it can be concluded that the pseudolinear regression performed significantly worse, resulting in both larger mean values and variances of the residuals.

To demonstrate the effect of the initial guess $\mathbf{g}_0$ on the convergence of the estimated parameters, in Fig 9 all four scenarios are considered according to Eq (44) what resulted in the convergence to one of the four symmetrical solutions defined by Eq (32).

By drastically reducing the number of samples to $N = 100$, we could obtain highly erroneous sample autocorrelation function $\hat{R}_{\epsilon\epsilon}(n)$ as shown in Fig 10. Evaluating criterion given by Eq (35) resulted in $12.8544 \not> 14.7992$, what implies that there is no solution to the estimation

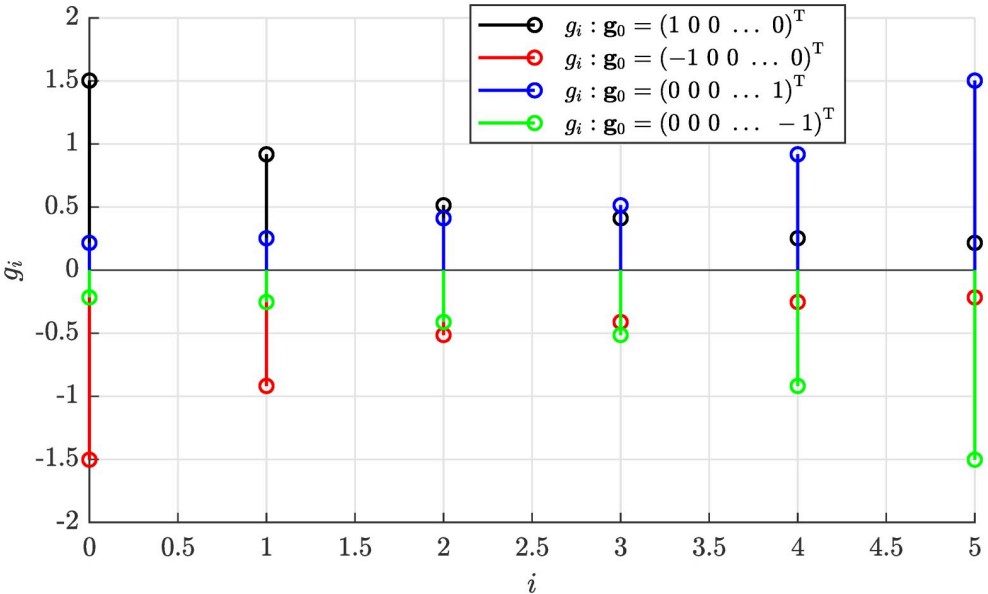

**Fig 9. Effect of the initial guess $\mathbf{g}_0$ on the convergence of the estimated parameters towards one of the four solutions.**

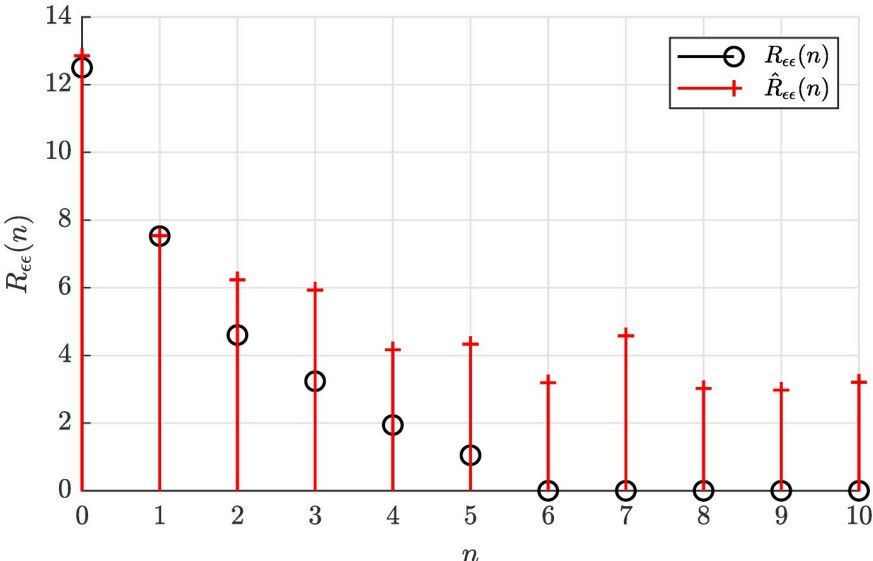

**Fig 10. Actual autocorrelation function $R_{\epsilon\epsilon}(n)$ and the estimated autocorrelation function $\hat{R}_{\epsilon\epsilon}(n)$ with $N = 100$.**

problem. This claim was also confirmed by attempting to solve the corresponding nonlinear equation system given by Eq (29) numerically using the Newton-Raphson algorithm and the Levenberg–Marquardt algorithm what resulted in a stalled convergence as documented in Fig 11.

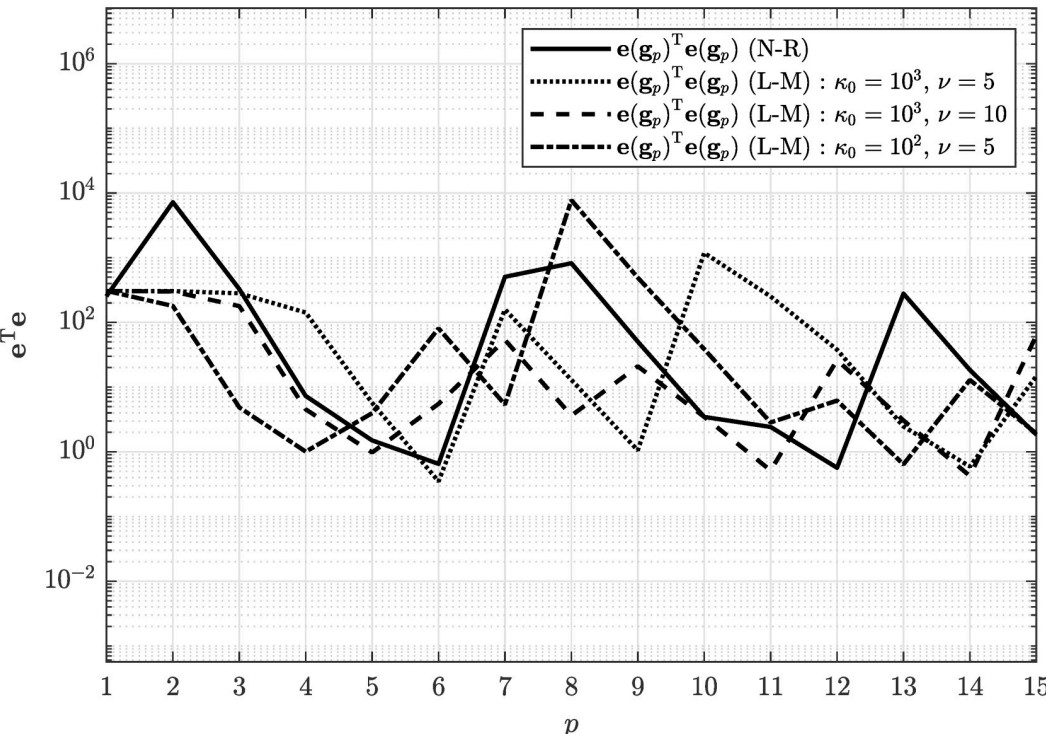

**Fig 11. Non-converging numerical solution of the non-linear equation system through the iterations $p$ using the Newton-Raphson algorithm and the Levenberg–Marquardt algorithm.**

The results presented above clearly indicate that the accuracy of the parameter estimate as well as the existence of the solution depend on the number of processed samples $N$, which determines the accuracy of the sample autocorrelation function. It should be remarked that a lower number of samples $N$ does not necessarily imply that the estimation problem has no solution, while the presented example is purely for demonstration of such a scenario. However, the fact is that the higher the uncertainty of the sample autocorrelation function, the higher the chance of not having a solution of the estimation problem.

## Non-stationary processes

We will continue the experimenting by performing the online parameter estimation of a non-stationary process in order to demonstrate the adaptive features of the proposed identification method. Consider a non-stationary moving average process with a step-wise (discontinuous) transition of parameters given as

$$\epsilon_{(k)} = (1 + 0.6z^{-1} + 0.5z^{-2} + 0.4z^{-3} + 0.2z^{-4})\eta_{(k)} \quad \forall k < 20000 \ ,$$
$$\epsilon_{(k)} = (1 + 0.7z^{-1} + 0.6z^{-2} + 0.3z^{-3} + 0.1z^{-4})\eta_{(k)} \quad \forall k \geq 20000 \ . \tag{56}$$

Online estimation of the aforementioned non-stationary moving average process was performed by solving Eqs (48) and (49) based on the recursively updated sample autocorrelation function $\hat{R}_{\epsilon\epsilon}$ according to Eq (27) while processing the sequence of $N = 40000$ samples and considering the forgetting factor $\lambda = 0.9998$. The corresponding timeseries of $\eta$ is in S4 Dataset.

The proposed online autocorrelation matching method will be compared with the recursive pseudolinear regression approach [6, 10, 11]. It is important to remark that the RPLR is the only relevant method capable of estimating non-stationary moving average processes online with a comparable computational complexity. The RPLR is derived from the recursive least squares method [6], which has the iteration given by equations

$$Y(k) = \frac{P(k-1)h(k)}{\lambda + h^{\mathrm{T}}(k)P(k-1)h(k)} \ , \tag{57a}$$

$$P(k) = \frac{1}{\lambda}\left(P(k-1) - Y(k)h^{\mathrm{T}}(k)P(k-1)\right) \ , \tag{57b}$$

$$\hat{\theta}(k) = \hat{\theta}(k-1) + Y(k)(\epsilon_{(k)} - h^{\mathrm{T}}(k)\hat{\theta}(k-1)) \ , \tag{57c}$$

where $h \in \mathbb{R}^{n_g}$ is the regression vector, $Y \in \mathbb{R}^{n_g}$ is the correction vector, $P \in \mathbb{R}^{n_g \times n_g}$ is the covariance matrix of the parameter estimate, $\hat{\theta} \in \mathbb{R}^{n_g}$ is the vector of estimated parameters and $\lambda$ is the forgetting factor.

The parameter vector gets

$$\hat{\theta} = \left(\begin{matrix} \hat{g}_1 & \hat{g}_2 & \cdots \hat{g}_{n_g} \end{matrix}\right)^{\mathrm{T}} . \tag{58}$$

These equations are supplemented by the vital formula for estimating the input noise as

$$\hat{\eta}_{(k)} = \epsilon_{(k)} - h^{\mathrm{T}}(k)\hat{\theta}(k) \ . \tag{59}$$

The regression vector $h(k)$ will be formed by the estimated input noise as

$$h(k) = \begin{pmatrix} \hat{\eta}_{(k-1)} & \hat{\eta}_{(k-2)} & \cdots \hat{\eta}_{(k-n_g)} \end{pmatrix}^{\mathrm{T}}. \tag{60}$$

Note that in the case of RPLR, the estimated model polynomial is monic, i.e. $g_0 = 1$, hence this method provides only the $n_g$ remaining coefficients.

Based on the equations above, the following conclusions can be drawn about the computational complexity per iteration and the overall memory footprint of the RPLR. To store $P(k)$, $P(k-1)$, $Y(k)$, $\hat{\theta}(k)$, $h(k)$ there are required at least $2n_g^2 + 3n_g$ real-valued variables. It is a known characteristic of the recursive least squares method that not all updates given by Eq (57) can be performed "in place" without requiring an additional memory allocation for temporary variables and intermediate calculations, so the actual memory footprint is even larger. Therefore, it is apparent that the total memory footprint exceeds that of the proposed method, which was analyzed and discussed earlier.

Concerning the computational complexity of the recursive least squares update given by Eq (57), linear operations including the matrix-vector multiplications, inner and outer products, as well as scalar multiplications and element-wise matrix additions are required. Due to multiple matrix operations with quadratic time complexity, the resulting complexity will be higher than that of the correlation-based method, which was shown to have only one quadratic term in the complexity. However, considering big O notation, the asymptotic computational complexity of this algorithm is still quadratic $\mathcal{O}(n_g^2)$.

In the experiment, we will consider the initial parameter estimate $\theta(0) = \mathbf{0}$, the initial covariance matrix of the estimate $P(0) = I \times 10^{10}$ and the same forgetting factor $\lambda = 0.9998$ to achieve the fairness of the comparison. The evolution of the estimated model parameters with respect to the number of processed samples $k$ of the correlated noise sequence $\epsilon$ is shown in Fig 12. This figure documents that the model parameters can effectively adapt due to the time-varying nature of the non-stationary moving average process. The achieved accuracy of the parameter estimate can be considered satisfactory, while it rapidly improves with an increasing number of processed samples $k$. The adaptation due to the step-wise change of the process parameters occurring at $k = 20000$ required approximately 10000 more samples for the estimated parameters to approach the actual values. It can be concluded that the performance of both compared methods is satisfactory and quite similar. The evolution of sample autocorrelation function $\hat{R}_{\epsilon\epsilon}(n, k)$ estimated online according to recursive formula given by Eq (27) is plotted in Fig 13.

In practice, the adaptation rate can be tuned to achieve the desired behavior by modifying the forgetting factor $\lambda$. The above experiment was repeated first with a lower forgetting factor $\lambda = 0.999$, which implies stronger forgetting of older samples, resulting in a faster adaptation rate. Second, a higher forgetting factor $\lambda = 0.99995$ was considered, implying weaker forgetting of older samples and resulting in a slower adaptation rate. Fig 14 then shows that the adaptation of parameters was faster than in Fig 12, but their volatility (uncertainty) also increased significantly, which is undesired. In contrast, Fig 15 demonstrates slower and less volatile adaptation of the estimated parameters.

The online estimation of the non-stationary process given by Eq (56) was repeated for multiple ($M$) different realizations of the random input white noise sequences $\eta$. The corresponding timeseries of $\eta$ are in S5 Dataset. For the $i$-th of the total $M = 100$ realizations, the parameters were determined as functions of the sample number $k$ using the proposed correlation method and the RPLR, as described above. This statistical approach mitigates the influence of a realization of the input noise sequence $\eta$ on the parameter estimate, thereby enabling a more robust assessment of the estimation performance. To this end, the mean values of the

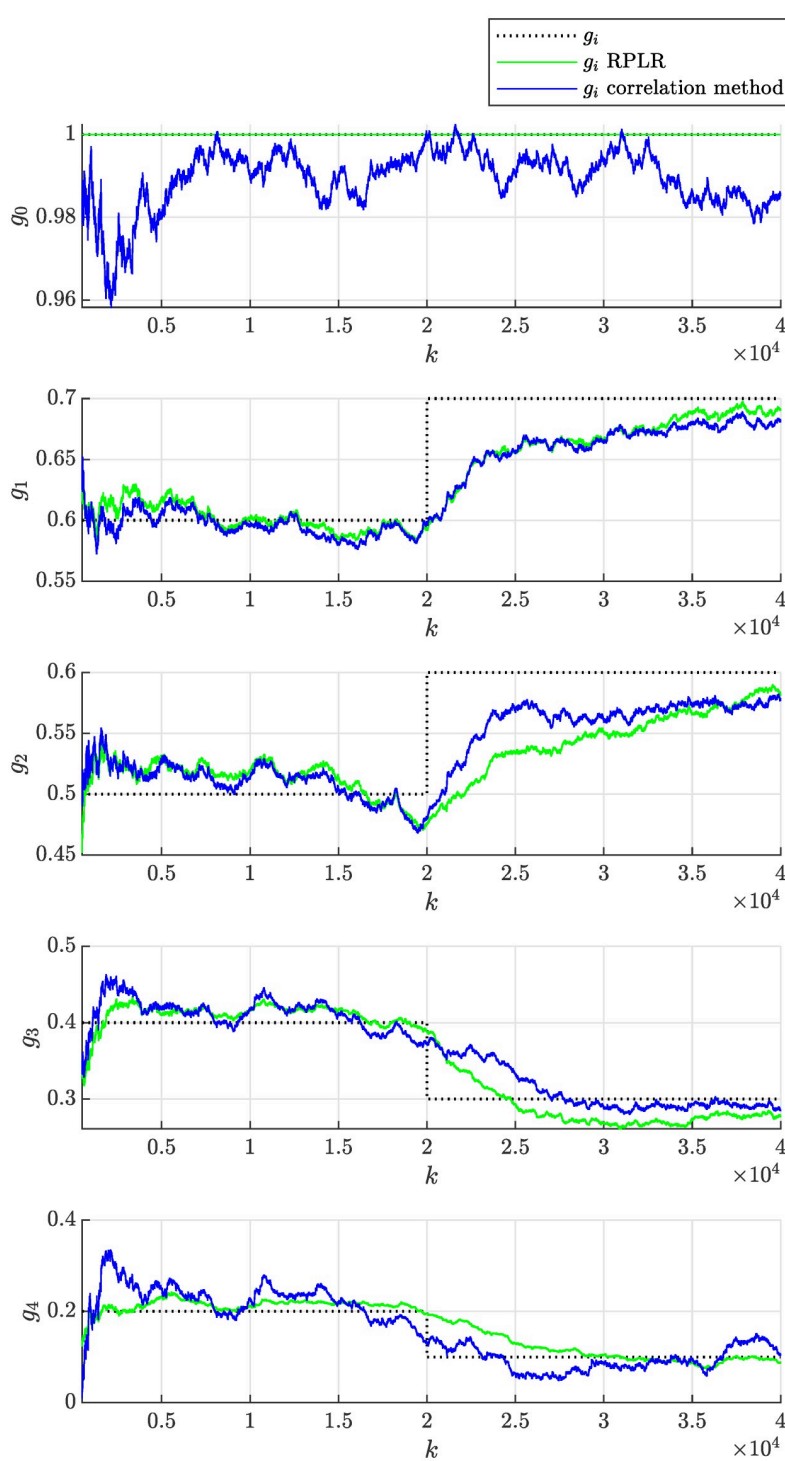

**Fig 12. Online estimation of the coefficients $g_i(k)$ of the non-stationary moving average process as a function sample number $k$ obtained assuming $\lambda$ = 0.9998 using the proposed correlation method and the RPLR.**

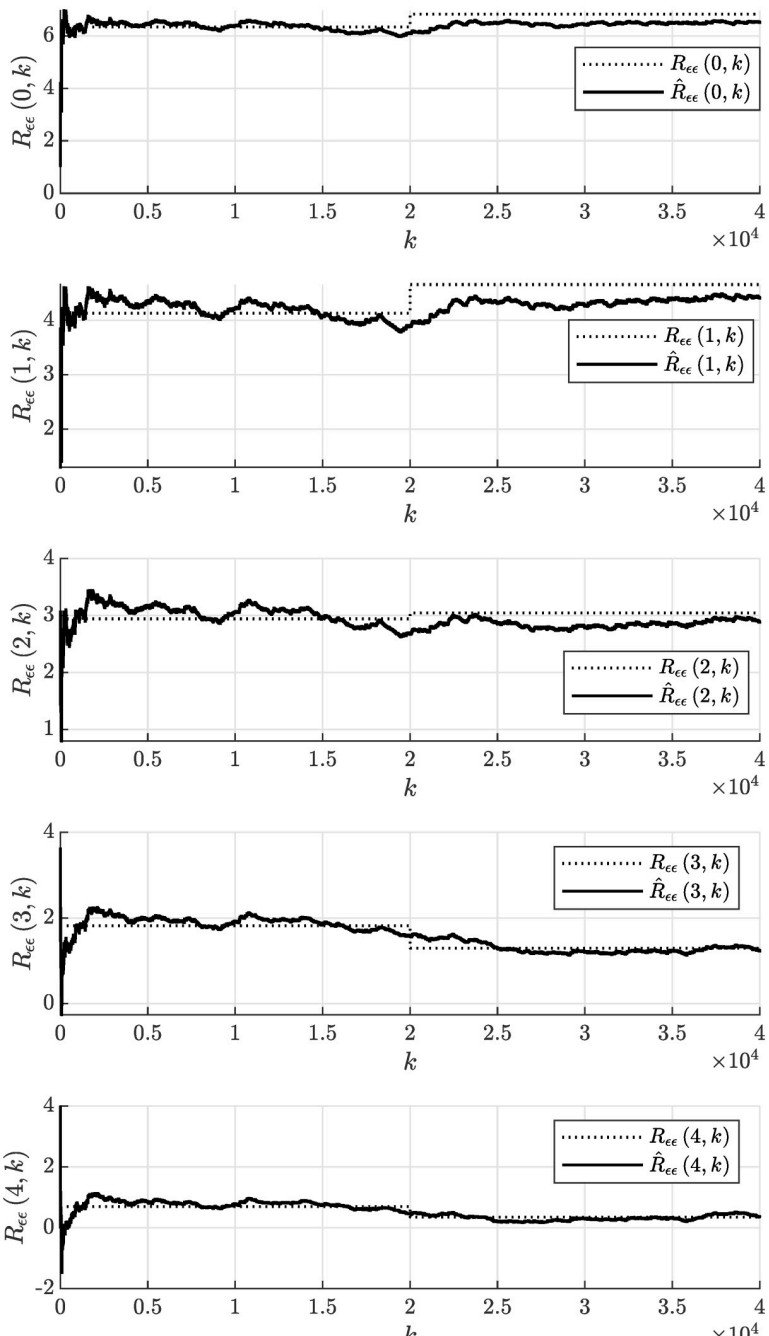

**Fig 13. Actual autocorrelation function $R_{\epsilon\epsilon}(n, k)$ and sample autocorrelation function $\hat{R}_{\epsilon\epsilon}(n, k)$ estimated online assuming $\lambda = 0.9998$ in the case of non-stationary process.**

estimated parameters as functions of the sample number $k$ were evaluated across $M$ realizations of the input noise sequence and are plotted in Fig 16. This figure shows that the proposed correlation method provides surprisingly fast initial adaptation and, more importantly, the estimate converges to the actual parameters with little to no statistical bias, compared to the RPLR, which appears to have significant bias of $g_3$ and $g_4$.

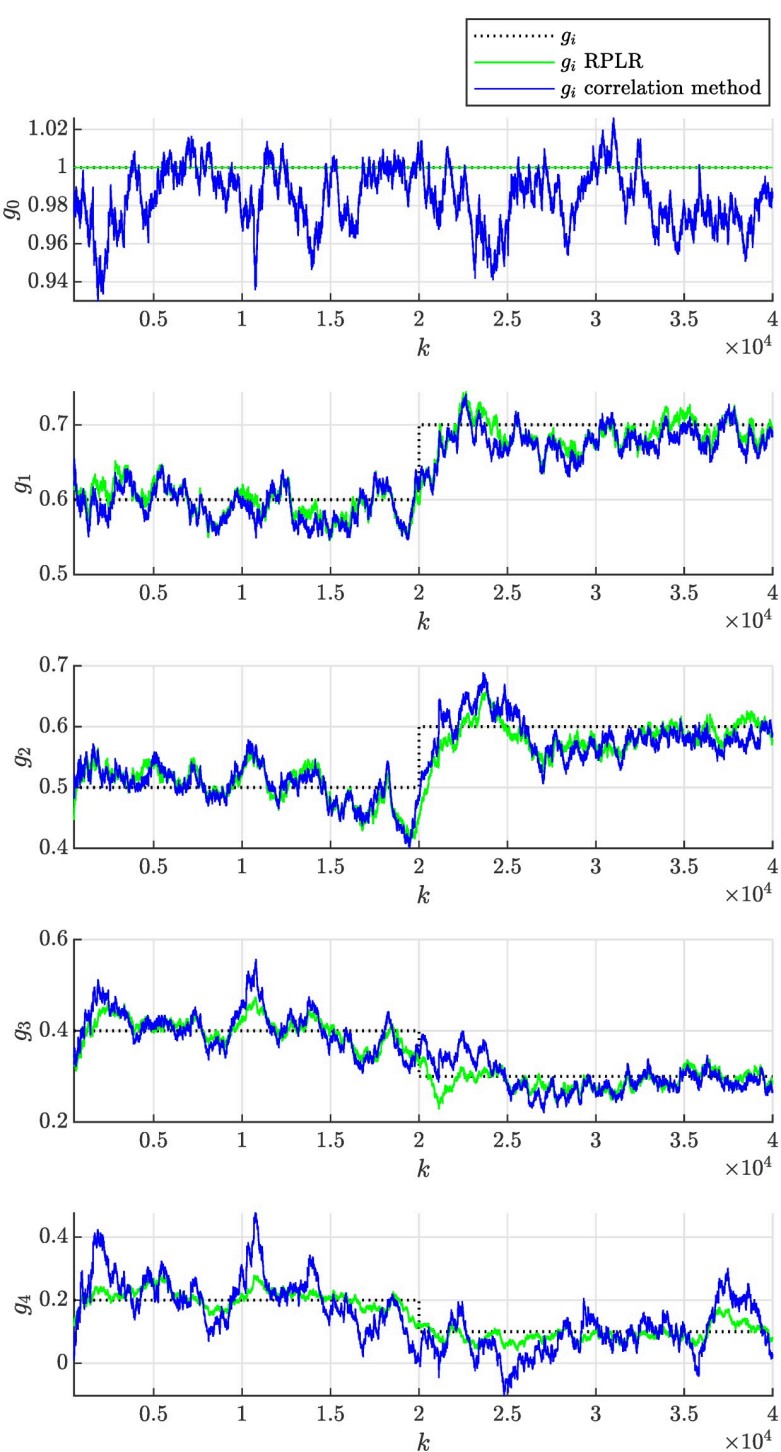

**Fig 14. Online estimation of the coefficients $g_i(k)$ of the non-stationary moving average process as a function of sample number $k$ obtained assuming $\lambda = 0.999$ using the proposed correlation method and the RPLR.**

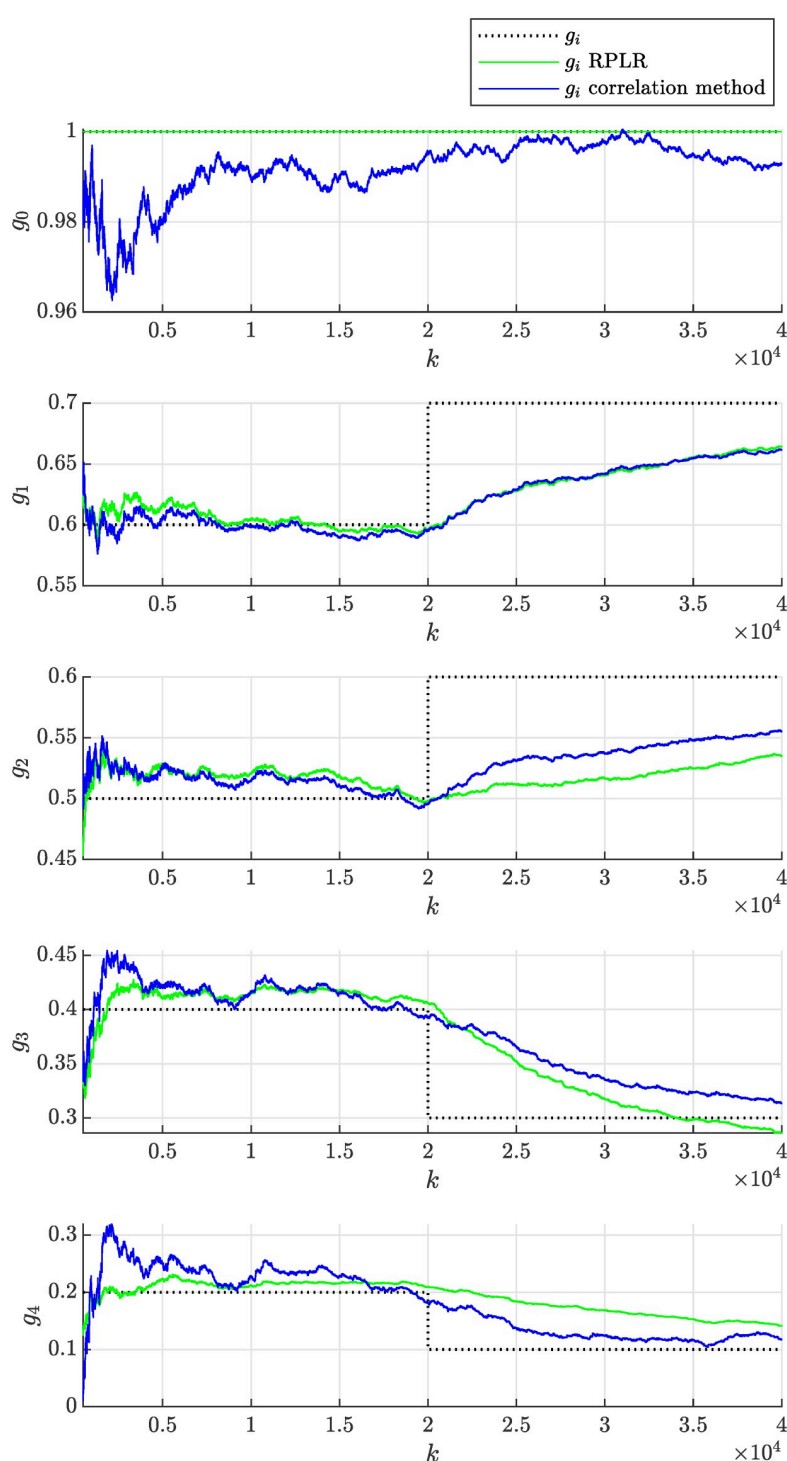

**Fig 15. Online estimation of the coefficients $g_i(k)$ of the non-stationary moving average process as a function of sample number $k$ obtained assuming $\lambda = 0.99995$ using the proposed correlation method and the RPLR.**

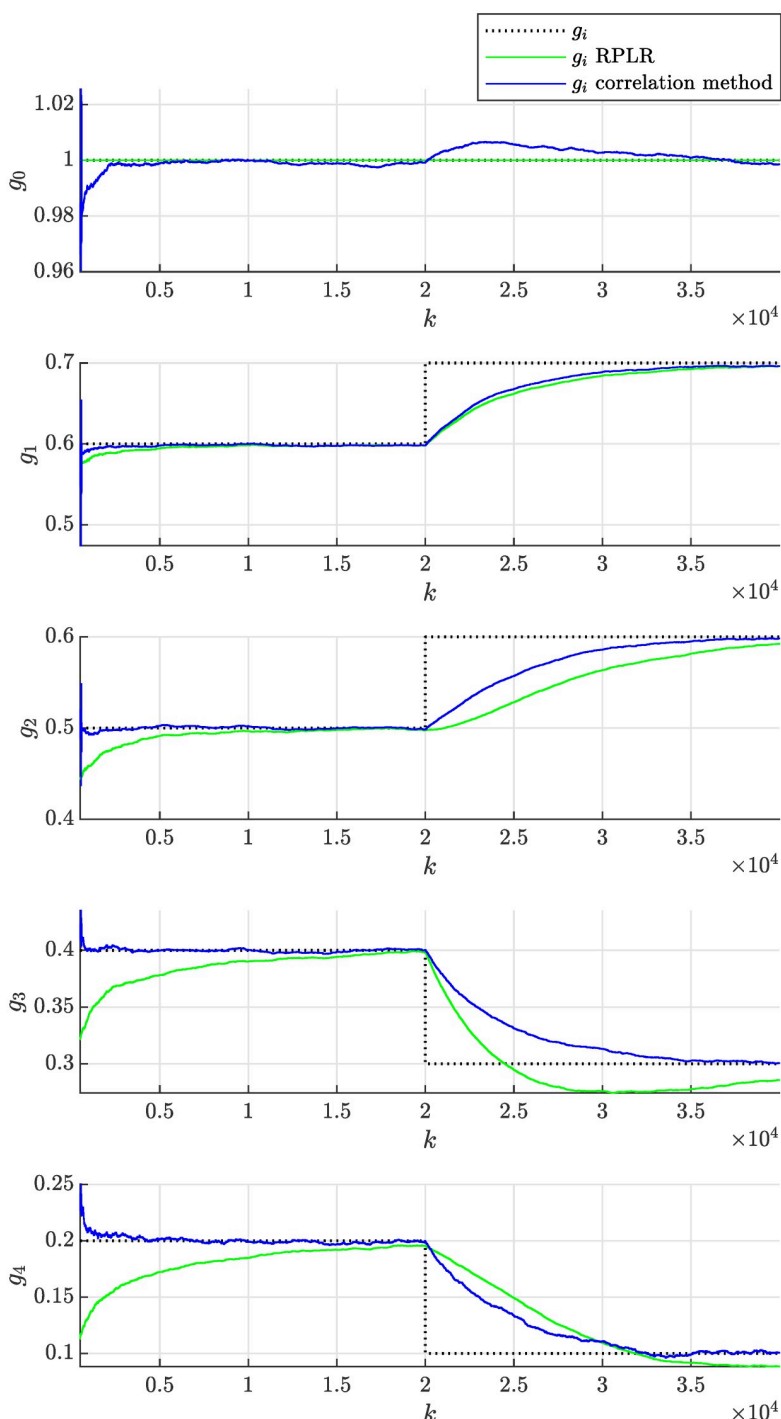

**Fig 16. Mean values of the estimated of the coefficients $g_i(k)$ of the non-stationary moving average process as a function of sample number $k$ obtained assuming $\lambda = 0.99998$ using the proposed correlation method and the RPLR across $M = 100$ realizations of input noise sequence.**

To demonstrate that the proposed estimation algorithm can reliably handle real-world data in the form of sampled colored noise sequences, rather than only artificially generated ones as presented so far, further experimentation will be conducted. Real data involving a correlated (colored) noise sequence, which represents a realization of some non-stationary moving average process with unknown coefficients, typically contain adverse features such as nonlinearities, varying noise levels, and nonzero mean that may impact parameter estimation. However, it should be noted that the actual parameters of the underlying moving average process are completely unknown; therefore, directly assessing the accuracy will not be possible.

For this experiment, we will work with the publicly available Environmental Sound Classification dataset (ESC-50) [29]. The ESC-50 dataset is a labeled collection of audio recordings originally designed for testing methods of sound classification, spectral analysis, and signal processing in general. It comprises 2000 five-second clips across 50 different classes, including natural, human, and domestic sounds. The most relevant types of sounds for this experiment, which naturally exhibit the characteristics of non-stationary colored noise, are wind, sea waves or rainfall. These natural noise sounds have time-dependent spectra and non-stationary statistical properties that evolve throughout the duration of the sampled sound sequence. The practical aim of the experiment is to determine the evolution of the coefficients of the underlying moving average process that produces this sound, i.e., to determine the mathematical model of the noise. The database of samples was downloaded from the publicly available repository https://github.com/karolpiczak/ESC-50.

The considered sampling frequency $f_s$ = 22500 Hz and the duration of each recording for 5 seconds implies the total number of samples $N$ = 110250. For the colored noise from the recording *2–81731-A-10.wav* (rainfall), the model order was chosen as $n_g$ = 5 based on the prior assessment of the autocorrelation function, and the forgetting factor $\lambda$ was carefully tuned to obtain the optimal adaptation rate as $\lambda$ = 0.9995. The envelope of the colored noise signal $\epsilon_{(k)}$ from the recording *2–81731-A-10.wav* visualized in Fig 17 suggest that this signal features highly variable noise levels.

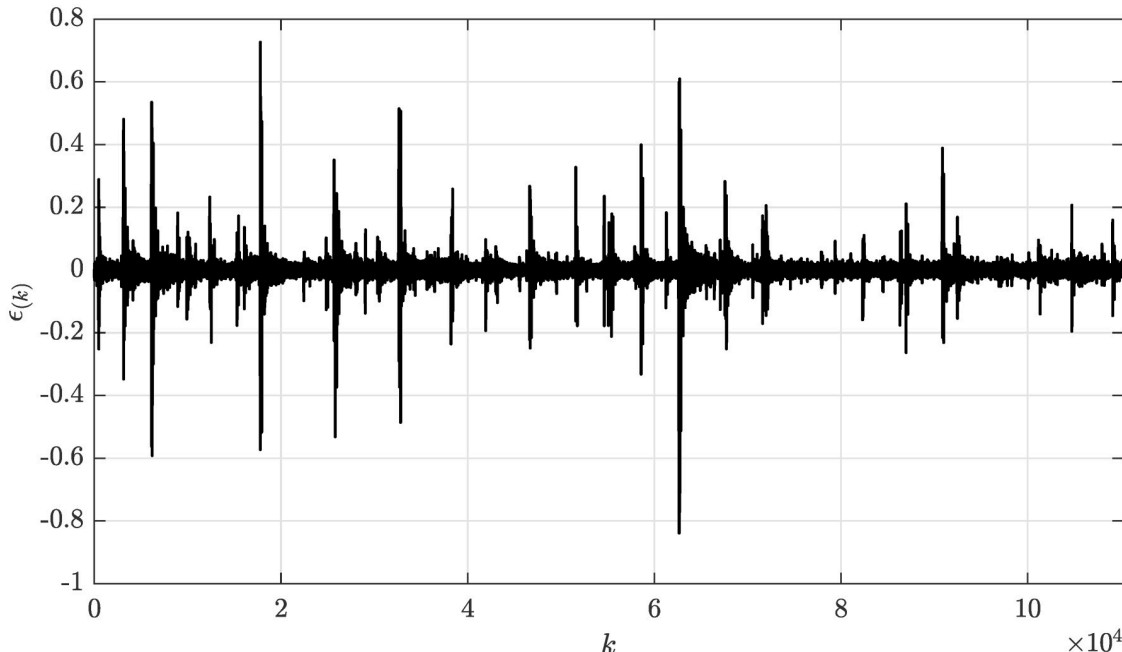

**Fig 17. Envelope of the colored noise signal $\epsilon_{(k)}$ from recording *2–81731-A-10.wav* as a function of sample number *k*.**

The resulting evolution of the estimated model parameters with respect to the sample number $k$ using the proposed correlation method and the RPLR is shown in Fig 18. This figure documents relatively large and abrupt fluctuations of the parameters, while this primarily correlates with the volatility of the signal amplitude (envelope), which translated into

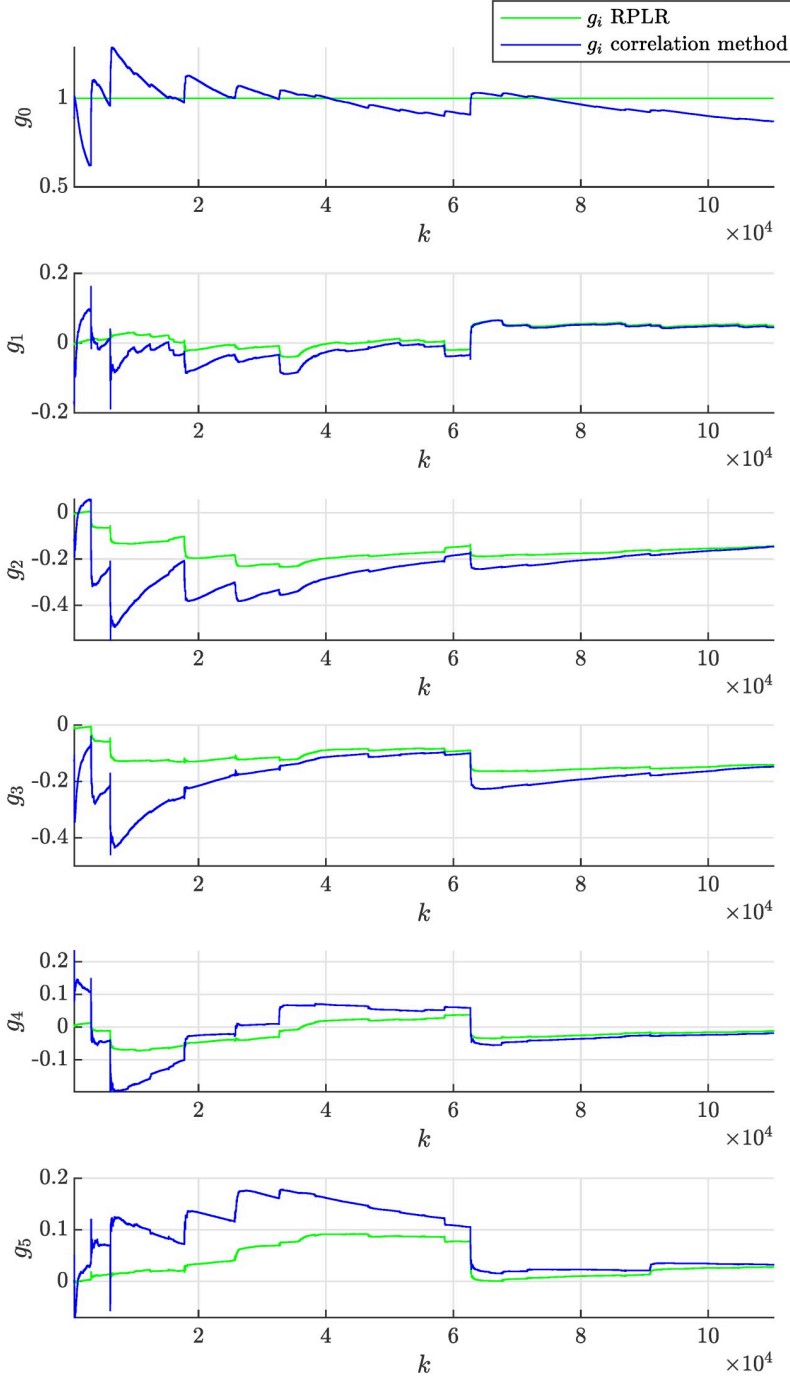

**Fig 18. Online estimation of the coefficients $g_i(k)$ of the non-stationary moving average process based on the colored noise data from recording *2–81731-A-10.wav* as a function of sample number $k$ obtained using the proposed correlation method and the recursive pseudolinear regression approach.**

simultaneous change of all model coefficients. It is also noticeable that the RPLR provided different results, as it is not capable of estimating models with varying static gain (variable noise levels) because the method assumes the first model coefficient is fixed as $g_0 = 1$.

Finally, a quantitative benchmarking experiment will be conducted to compare the computational efforts required for a single iteration of the proposed correlation method and the recursive pseudolinear regression method in the case of online estimation of non-stationary processes. To ensure a fair comparison when measuring the execution time of a single iteration (runtime), we will use a dedicated Matlab function, *timeit*, which runs the tested functions multiple times automatically to provide a more stable and averaged measurement of the execution time. This helps reduce the impact of anomalies, such as background processes consuming CPU resources, during the benchmark. It is important to note that careful implementation of both methods was vital to avoid potentially biased results that could favor one method over the other only due to slight programming differences affecting the overall performance.

The resulting dependency of the average execution time per iteration for the compared methods, as a function of the number of estimated parameters $n_g$ (the order of the moving average model), is shown in Fig 19. This figure demonstrates that the recursive pseudolinear regression method is slower than the correlation-based method, which provides at least 2x faster iteration execution. This experiment also confirms that both the proposed correlation method and the recursive pseudolinear regression have quadratic asymptotic time complexity; however, the latter has a larger constant growth factor (gain). It is important to note that the exact ratio between these complexities (runtimes) also depends on the programming language, compiler, and hardware platform. Nevertheless, the correlation-based method will generally

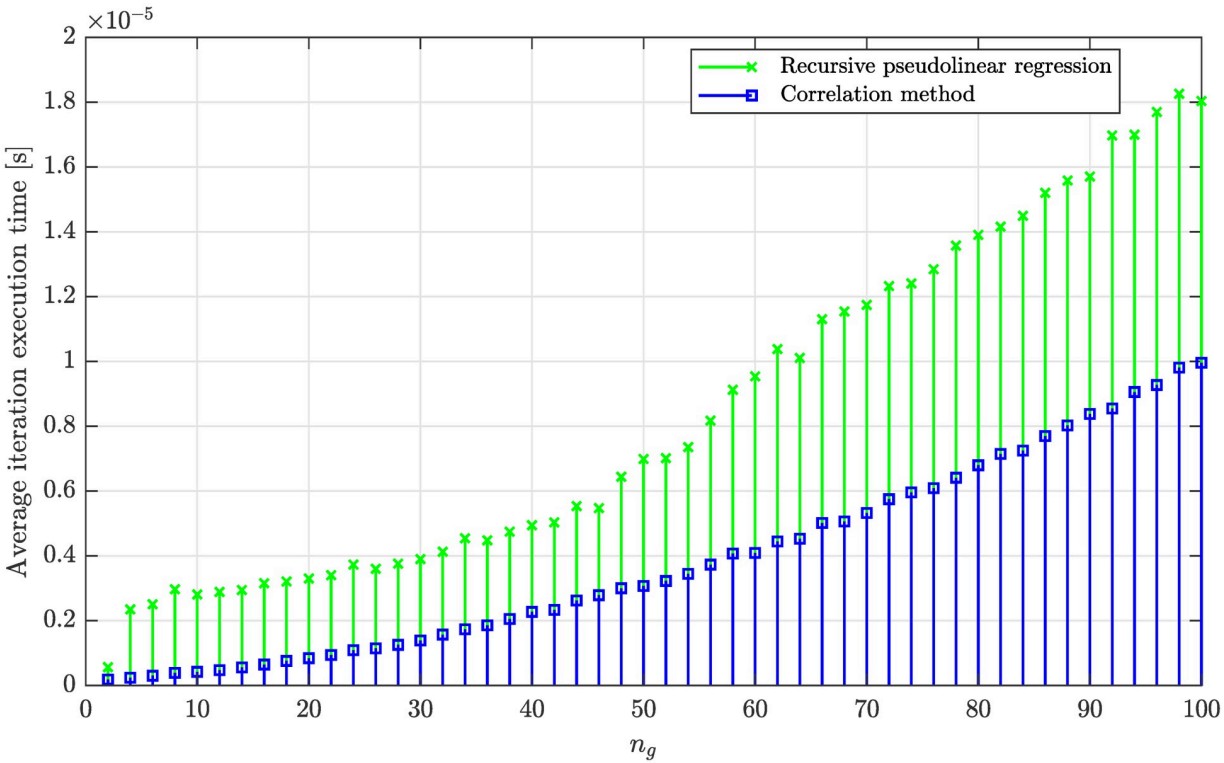

**Fig 19. Dependency of the average execution time of a single iteration corresponding to the proposed correlation method and the recursive pseudolinear regression as a function of the number of estimated parameters $n_g$.**

outperform the recursive pseudolinear regression due to the lower number of operations with quadratic time complexity and reduced nested loop resulting from back substitution given by Eq (48).

## Conclusion

The paper proposed a novel method for estimating stationary and non-stationary moving average processes based on matching the autocorrelation function of the model output to the sample autocorrelation function of a colored noise sequence (time series). The parameter estimation problem in the case of stationary processes resulted in a nonlinear equation system involving quadratic terms of the unknown coefficients. In order to find a solution of this equation system, the residual function was introduced and its root was determined numerically using the Newton-Raphson and the Levenberg-Marquardt algorithms. We also studied the existence of possible solutions of the estimation problem and thus the feasibility of its numerical approximation. It was demonstrated that there exist four symmetrical solutions satisfying the nonlinear equation system while deriving the necessary condition for their existence. There was also studied the effect of initial guess of the estimated parameters in the Newton-Raphson and the Levenberg-Marquardt algorithm on the convergence of the numerical solution towards one of the four possible solutions. The existence of the multiple solution and the effect of initial guess on the convergence was confirmed through the experiment.

However, the most significant contribution is an extension of this technique to online estimation of non-stationary moving average processes. Concerning the online estimation, we proposed a new recursive formula to effectively update the sample autocorrelation function while considering the exponential weighting. This turned out to be crucial for estimating non-stationary processes, where the exponential forgetting of older samples of the dataset is necessary to ensure adaptation of the estimated parameters. Besides that, this recursive formula allowed for a significant reduction in the online computation burden and minimized the memory footprint. The estimation problem of non-stationary processes resulted in the formulation of one nonlinear equation and $n_g$ linear equations with respect to the estimated coefficients, which was shown to be efficiently solved by the backward substitution due to the triangular structure of its matrix. This allowed to drastically minimize the number of calculations to be carried out online at each sample. We demonstrated that the algorithm has quadratic per-iteration complexity. We also showed that the matrix of this linear equation system is formed by the model coefficients estimated from the previous iterations, hence the estimation algorithm is recursive in its nature.

In contrast to other correlation-based methods published in the literature, the proposed approach is not based on the minimization of the autocorrelation function matching scalar criterion and the solution of the corresponding nonlinear least squares problem, but rather on the numerical root finding strategy applied to multivariate vector function of the residuals. Compared with the recursive pseudolinear regression, which assumes monic polynomial of the model, the proposed method allows to estimate more general models and processes with variable variance of the input noise.

The presented experiments confirmed that the proposed method can provide estimates of stationary and non-stationary moving average processes with the accuracy comparable, or even superior to the conventional prediction-error methods. The experimenting involved a comparison of the proposed method with the method of Durbin, pseudolinear regression and its recursive variant. Validating the estimation method on a specific non-stationary process defined by a step-wise change of the parameters resulted in no statistical bias and relatively fast model adaptation, which was shown to be tunable by adjusting the forgetting factor λ.

The statistical evaluation, considering multiple realizations of the input noise sequence, demonstrated that the correlation-based method performed similarly to Durbin's method and significantly outperformed the pseudolinear regression in terms of statistical bias and efficiency. It can be concluded that the accuracy of the parameter estimate is related to the variance of the sample autocorrelation function, which is determined by the number of processed samples and by the forgetting factor.

Given these points, the proposed correlation method emerges as a compelling alternative to the traditional prediction-error methods as it offers comparable or even better estimate accuracy and slightly lower computational complexity.

Despite the advantages, several limitations and shortcomings have been identified that need to be acknowledged. While we analyzed the impact of the forgetting factor $\lambda$ on the parameter estimate, determining the optimal $\lambda$ in practice can be challenging, especially in problems where the statistical properties change rapidly. The existence of four symmetrical solutions to the estimation problem means that four different models share the same autocorrelation function, which may lead to ambiguity. The convergence of the algorithm is sensitive to the choice of the initial guess; a poor choice of the initial guess may result in slow convergence or convergence to a symmetrical solution. Finally, the derived feasibility condition means that for an improper sample autocorrelation function, which may be a result of insufficient data quantity, the estimation problem might not have a solution.

These findings suggest the following possible directions for the future research. Future research should explore ways to reliably select the correct solution from the set of possible symmetrical solutions. This could involve applying regularization techniques that favor specific solutions based on prior knowledge or model assumptions. Another option is implementing an adaptive forgetting factor $\lambda$ that automatically adjusts based on the data's characteristics, which could improve the performance of the method.

The findings presented in this paper can be adopted and may also have an impact in other problems related to statistics and engineering where it is required to estimate the coefficients of a moving average process. These involve forecasting of time series in specific economic and financial applications [3], such as commodity [30] and stock market prices [31].

## Supporting information

**S1 Dataset. Time series $\eta$ for stationary process, $N$ = 5000 samples.**
(TXT)

**S2 Dataset. Time series $\eta$ for stationary process, $N$ = 500 samples.**
(TXT)

**S3 Dataset. $M$ = 100 realizations of time series $\eta$ for stationary process, $N$ = 5000 samples.**
(TXT)

**S4 Dataset. Time series $\eta$ for non-stationary process, $N$ = 40000 samples.**
(TXT)

**S5 Dataset. $M$ = 100 realizations of time series $\eta$ for non-stationary process, $N$ = 40000 samples.**
(TXT)

## Author Contributions

**Conceptualization:** Martin Dodek.

**Formal analysis:** Martin Dodek, Eva Miklovičová.

**Funding acquisition:** Martin Dodek.

**Investigation:** Martin Dodek, Eva Miklovičová.

**Project administration:** Martin Dodek.

**Software:** Martin Dodek.

**Validation:** Martin Dodek, Eva Miklovičová.

**Writing – original draft:** Martin Dodek.

**Writing – review & editing:** Eva Miklovičová.

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
