## [Decision Letter · Decision Letter 0]

2 Oct 2024

PONE-D-24-28946Estimation of moving average processes using novel autocorrelation function matching method extended to online estimation of non-stationary processesPLOS ONE

Dear Dr. Dodek,

Thank you for submitting your manuscript to PLOS ONE. After careful consideration, we feel that it has merit but does not fully meet PLOS ONE’s publication criteria as it currently stands. Therefore, we invite you to submit a revised version of the manuscript that addresses the points raised during the review process.

We look forward to receiving your revised manuscript.

Kind regards,

Arne Johannssen

Academic Editor

PLOS ONE

Journal Requirements:

3. Thank you for stating the following financial disclosure: “This work was supported by the call for PhD students and young researchers of Slovak University of Technology in Bratislava to start a research career (Grant 23-03-01-B Impulsive Control of Biosystems).

Call for PhD students and young researchers was supported by call 09I03-03-V05, financed by RRP SR.”

4. We note that your Data Availability Statement is currently as follows: “All relevant data are within the manuscript and in Supporting Information files.”

Please confirm at this time whether or not your submission contains all raw data required to replicate the results of your study. Authors must share the “minimal data set” for their submission. PLOS defines the minimal data set to consist of the data required to replicate all study findings reported in the article, as well as related metadata and methods (https://journals.plos.org/plosone/s/data-availability#loc-minimal-data-set-definition). For example, authors should submit the following data: - The values behind the means, standard deviations and other measures reported; - The values used to build graphs; - The points extracted from images for analysis. Authors do not need to submit their entire data set if only a portion of the data was used in the reported study. If your submission does not contain these data, please either upload them as Supporting Information files or deposit them to a stable, public repository and provide us with the relevant URLs, DOIs, or accession numbers. For a list of recommended repositories, please see https://journals.plos.org/plosone/s/recommended-repositories. If there are ethical or legal restrictions on sharing a de-identified data set, please explain them in detail (e.g., data contain potentially sensitive information, data are owned by a third-party organization, etc.) and who has imposed them (e.g., an ethics committee). Please also provide contact information for a data access committee, ethics committee, or other institutional body to which data requests may be sent. If data are owned by a third party, please indicate how others may request data access.

5. Please remove your figures from within your manuscript file, leaving only the individual TIFF/EPS image files, uploaded separately. These will be automatically included in the reviewers’ PDF**.**

7. We are unable to open your Supporting Information file data_offline_N500.mat, data_offline_N5000.mat and data_online.mat. Please kindly revise as necessary and re-upload.

Additional Editor Comments:

**Please carefully address the issues raised by the reviewers before the paper can be reconsidered for potential publication in PONE.**

Reviewers' comments:

Reviewer's Responses to Questions

**Comments to the Author**

1. Is the manuscript technically sound, and do the data support the conclusions?

Reviewer #1: Yes

Reviewer #2: Partly

2. Has the statistical analysis been performed appropriately and rigorously? 

Reviewer #1: Yes

Reviewer #2: No

3. Have the authors made all data underlying the findings in their manuscript fully available?

Reviewer #1: Yes

Reviewer #2: Yes

4. Is the manuscript presented in an intelligible fashion and written in standard English?

Reviewer #1: Yes

Reviewer #2: Yes

5. Review Comments to the Author

Reviewer #1: The manuscript is well-organized and written, here are some recommendations to consider

1. The title is dense, it needs simplification or rephrasing to enhance clarity and impact.

2. The novelty of the approach and key findings need to be emphasized in the abstract.

3. While various methods are mentioned in the introduction section, the motivation for choosing the specific approach should be highlighted more explicitly.

4. Clearly state the assumptions made in the derivations (e.g., ergodicity, properties of the input noise).

5. Consider discussing how the choice of the forgetting factor (λ) affects the results and any guidelines for selecting it.

6. Discuss some limitations or potential areas for future research to provide a more balanced view.

Reviewer #2: One of the main concerns with the article is the lack of sufficient experimental validation to support the claim that the proposed method is suitable for real-time applications. Although the capability for online estimation is mentioned, there are no robust experiments demonstrating that the algorithm can operate in real-time scenarios with latency constraints and continuous processing. The authors should include experiments using real-time data to analyze the method's ability to adapt to rapid parameter changes under real-world, real-time conditions.

This leads to a second observation: the proposed method has been tested primarily in controlled environments and simulations with synthetically generated data. There is no evaluation of how the method behaves in the presence of real-world noise or under less ideal conditions common in practical applications.

It is recommended to:

- Include tests with real datasets that introduce noise, missing data, or nonlinearities that could affect parameter estimation.

- Conduct a sensitivity analysis of the method with respect to different levels of noise and data variations to demonstrate that the algorithm is stable and capable of handling real-world situations.

Furthermore, the article makes strong claims about the reduction of computational complexity of the methodology, especially in the context of online estimation of non-stationary processes. However, a detailed analysis of the algorithm's time and space complexity is not provided, nor is there a quantitative comparison with other methods. It is advisable to compare the performance of the proposed method with other techniques, such as pseudolinear regression and prediction error-based methods, through quantitative benchmarking experiments.

Finally, although the proposed method is shown to outperform other methods in terms of estimation accuracy, the inclusion of additional statistical tests—such as hypothesis tests or significance analyses—would help to more rigorously validate that the observed differences between methods are statistically significant and not simply a product of data variability.

6. PLOS authors have the option to publish the peer review history of their article (what does this mean?). If published, this will include your full peer review and any attached files.

Reviewer #1: No

Reviewer #2: No

---

## [Author Response · Author response to Decision Letter 0]

18 Oct 2024

The respond to reviewers is elaborated in a separate file

See document Review response.

---

## [Decision Letter · Decision Letter 1]

5 Nov 2024

Estimation of stationary and non-stationary moving average processes in the correlation domain

PONE-D-24-28946R1

Dear Dr. Dodek,

We’re pleased to inform you that your manuscript has been judged scientifically suitable for publication and will be formally accepted for publication once it meets all outstanding technical requirements.

Kind regards,

Arne Johannssen

Academic Editor

PLOS ONE

Additional Editor Comments (optional):

Reviewers' comments:

Reviewer's Responses to Questions

**Comments to the Author**

1. If the authors have adequately addressed your comments raised in a previous round of review and you feel that this manuscript is now acceptable for publication, you may indicate that here to bypass the “Comments to the Author” section, enter your conflict of interest statement in the “Confidential to Editor” section, and submit your "Accept" recommendation.

Reviewer #1: All comments have been addressed

Reviewer #2: All comments have been addressed

2. Is the manuscript technically sound, and do the data support the conclusions?

Reviewer #1: Yes

Reviewer #2: Yes

3. Has the statistical analysis been performed appropriately and rigorously? 

Reviewer #1: Yes

Reviewer #2: Yes

4. Have the authors made all data underlying the findings in their manuscript fully available?

Reviewer #1: Yes

Reviewer #2: Yes

5. Is the manuscript presented in an intelligible fashion and written in standard English?

Reviewer #1: Yes

Reviewer #2: Yes

6. Review Comments to the Author

Reviewer #1: Thanks for all the modifications and taking care of the comments. just please make sure that the number and the referencing of equations are correct and the equations are well formatted.

Reviewer #2: I think that the revisions made fully address the concerns raised and enhance the quality of the manuscript. The additional experiments and detailed analysis provide greater clarity and confidence in the presented results. I am satisfied with the revision in this regard and consider that the text now meets the expected standards.

7. PLOS authors have the option to publish the peer review history of their article (what does this mean?). If published, this will include your full peer review and any attached files.

Reviewer #1: No

Reviewer #2: No

---

## [Editor Report · Acceptance letter]

17 Nov 2024

PONE-D-24-28946R1 

PLOS ONE

Dear Dr. Dodek, 

I'm pleased to inform you that your manuscript has been deemed suitable for publication in PLOS ONE. Congratulations! Your manuscript is now being handed over to our production team.

Kind regards, 

on behalf of

Profesor Arne Johannssen 

Academic Editor

PLOS ONE